

# Investigation of the ionospheric absorption response to flare events during the solar cycle 23 as seen by European and South African ionosondes

Veronika Barta[1], Gabriella Sátori [1], Kitti Alexandra Berényi[2, 1], Árpád Kis[1], Earle Williams[3]

[1] Geodetic and Geophysical institute, Research Centre for Astronomy and Earth Sciences, Sopron, Hungary
[2] Eötvös Loránd University, Budapest, Hungary
[3] Parsons Laboratory, Massachusetts Institute of Technology, Cambridge, USA

*Correspondence to*: Veronika Barta (bartav@ggki.hu)

**Abstract.** Systematic analysis of ionosopheric parameters measured at mid- and low-latitudes was performed to study the ionospheric response to solar flares. The lowest recorded ionosonde echo, the mimimum frequency (fmin, a qualitative proxy for the ''nondeviative'' radio wave absorption occurring in the D-layer), furthermore the dfmin parameter (difference between the value of the fmin and the mean fmin for reference days) have been investigated. The time series of the fmin and dfmin parameters recorded at meridionally-distributed ionosonde stations in Europe and South Africa were analyzed during eight X and M class solar flares during solar cycle 23. The solar zenith angles of the observation sites at the time of the selected flares have been also taken into account. Total and partial radio fade-out was experienced at every ionospheric stations during intense solar flares (> M6). The duration of the total radio fade-out varied between 15 and 150 min and it was highly dependent on the solar zenith angle of the ionospheric stations. Furthermore, a solar zenith angle-dependent enhancement of the fmin (2-9 MHz) and dfmin (1-8 MHz) parameters was observed at almost every stations. The fmin and dfmin parameters show an increasing trend with the enhancement of the X-ray flux. Based on the results, the dfmin parameter is a good qualitative measure for the relative variation of the "nondeviative" absorption especially in the case of the less intense solar flares which do not cause total radio fade-out in the ionosphere (class < M6).

## 1. Introduction

The most intense external forcing of the ionosphere from above is related to solar flares. These events are giant explosions on the surface of the Sun that suddenly release large amounts of electromagnetic energy at a wide range of wavelengths, particularly in the bands of X-radiation and extreme ultraviolet (EUV), for a very short duration (~ 30 minutes to ~ 1 hour, Tsurutani et al., 2009). Solar flares are classified as large (X), medium-size (M) and small (C) according to their peak flux (in watts per square meter, $Wm^{-2}$, $M \sim 10^{-5} - 10^{-4}$ $Wm^{-2}$, $X > 10^{-4}$ $Wm^{-2}$) of 0.1 to 0.8 nm X-rays near Earth, as measured on the GOES spacecraft. During solar flares, the suddenly increased radiation causes extra ionization of the neutral components in the sunlit hemisphere of the Earth's atmosphere over short time intervals (few minutes to 1 hour (Rishbeth and Garriot, 1969;



Tsurutani et al., 2009; Zolesi and Cander, 2014)). While hard X-rays (< 1 nm) penetrate deeply the ionosphere and could cause enhanced ionization in the D region during solar flares (Brasseur and Solomon, 1986; Rees, 1989; Hargreaves, 1992), the less energetic soft X-ray (1-10 nm) and far UV flux (80-102.6 nm) rather enhances the ionization in the E region (Rishbeth and Garriot, 1969). In addition to electromagnetic radiation, solar flares are also accompanied by energetic particles (protons and

electrons) with energies from some tens of keV to some hundreds of MeV, though they reach the Earth's atmosphere between a half and a few hours later, and cause impact ionization (Rishbeth and Garriot, 1969; Bothmer and Daglis, 2007; Tsurutani et al., 2009).

The approximate peak electron energy of a few keV causes a maximum ionization in the lower E region while during the so-called solar proton events (SPE) high energy protons (up to more than 100 MeV) cause ionization much deeper, in the D region

(Reid, 1986; Rees, 1989; Bothmer and Daglis, 2007). The significant enhancement of the electron density as a consequence of solar flares can create increased attenuation of electromagnetic waves propagating through the ionosphere. "*The mechanism of ionospheric radio wave absorption is well understood. Ionospheric electrons are accelerated by the electric field of the transiting radio wave. In the absence of collisions the electrons would simply reradiate the absorbed energy (with a phase lag because of their inertia). Because of the presence of the neutral atmosphere, the accelerated electrons suffer collisions with*

*the atmospheric constituents and incur an energy loss which results in a reduction of their reemitted signal*" (Sauer and Wilkinson, 2008). Since the atmospheric density, the collision frequency and also the recombination rate changes with altitude, the efficiency of radio wave absorption in the ionosphere strongly varies with altitude. The electron collision frequency is high in the D region ($2\times10^6$ s$^{-1}$) and the HF radio waves below 10 MHz can be strongly attenuated there (Zolesi and Cander, 2014). Thus, total radio fade-out for tens of minutes or hours can be the consequence of the enhancement of electron density caused

by increased radiation or energetic particles. Protons with energy less than 100 MeV will penetrate the Earth's atmosphere only at high latitudes (> 60°), causing radio wave fade-out, so called Polar Cap Absorption (PCA) events there (Bailey, 1964; Sauer and Wilkinson, 2008; Tsurutani et al., 2009). Generally, a PCA event begins a few hours after solar flares and lasts for some days (Rishbeth and Garriot, 1969; Zolesi and Cander, 2014). The loss of HF communication as a consequence of the enhanced absorption affects navigation systems, especially commercial aircraft operations. Thus the monitoring of the absorption and

D-, E-region electron density variation is an important issue from a practical point of view as well.

The principal method of observing PCA is through the use of a riometer (Relative Ionospheric Opacity Meter using Extra-Terrestrial Electromagnetic Radiation), which measures the absorption of the cosmic radio noise at a given high frequency, usually between 20 and 60 MHz at high latitudes (> 62° geodetic latitude in Europe) (Littlle and Leinbach, 1963; Stauning 1996). PCA occurs with considerable uniformity inside, as well as along the zone of maximum auroral activity (Bailey, 1964).

Furthermore, Rose and Ziauddin (1962) claimed that PCA can exist only at a geomagnetic latitude higher than 62°. In the study of Hargreaves and Birch (2005) it is found that the most effective bands are 9–40 MeV at 65 km and 40–80 MeV at 60 km. The model of Patterson et al. (2001) shows that the majority of the ionization resulting from the influx of solar energetic protons occurs in the altitude range from ~ 50-90 km but can extend to both higher and much lower altitudes depending upon the incoming energies and fluxes. The electron density (Ne) of the D region is enhanced by up to one order of magnitude down





to about 55 km prior to, during and after the solar proton event (SPE) on January 17, 2005. The largest Ne are found during the maximum of the X-ray flare on January 17. The electron density is still enhanced on January 18 when the X-ray flare decayed but the solar proton fluxes are still enhanced (Singer et al., 2011).

Enhanced X-ray fluxes during solar flares are known to cause increased ionization in the Earth's lower ionosphere (D region).
Sahai et al. (2006) have studied the 28 October 2003 solar flare event over the Brazilian sector using ionosonde data and found a lack of echoes in the ionograms for a 1 h period during the flare onset. They suggested that the reason for complete or partial radio signal fade-out could be intense absorption. The minimum frequency of reflection in radio soundings by ionosondes (fmin) depends on the absorption within the D region. The association between the solar flares and enhancement of fmin (> 100 %) in the ionosphere has been reported by Sharma et al. (2010). Zaalov et al. (2018) developed an empirical absorption
model using the combination of the Global Ionospheric Radio Observatory (GIRO, http://giro.uml.edu) data and ionogram modelling. More reliable and accurate evaluation of minimum frequency is possible thanks to their proposed method. The D region electron density (Ne) response to solar flares was studied with a medium frequency (MF) radar at Kunming (25.6°N, 103.8°E) (Li et al., 2018). They found a strong and positive correlation between Ne and the variation of X-rays during thirteen M class flares. Based on the results the Ne changes also depended on the onset time and duration of the flare. Furthermore, the
GNSS ground and satellite receivers offered further possibilities to study in high time (~ 30 s) and spatial resolution the solar flare effects on total electron content (TEC) (Afraimovich, 2000; Zhang et al., 2002; Tsurutani et al., 2005 and 2006). Tsurutani et al. (2009) summarized the "solar flare effects" on the ionosphere, and especially on TEC in a comprehensive review paper. In addition, during solar flares and high energy particle precipitation events, enhanced ionization of the D region can lower the reflection height of the VLF, ELF radio waveguide and perturb the amplitude of the propagating signals (Thomson and Clilverd,
2001; Thomson et al., 2004; Kolarski and Grubor, 2014). Guha et al. (2017) investigated solar flare effects on the D-region during 12 solar flares using a portable VLF station installed at Antarctica during summertime. They applied a Long Wave Propagation Capability (LWPC) model to study the daytime electron density changes during the flares and found an excellent correlation between the exponential fit of the modeled electron density change and the average X-ray flux change. Based on VLF measurements and TEC calculations Drakul et al. (2011) showed that the D-region's electron content (TECD)
contribution in TEC can reach several percent during solar flares. The effects of two extraordinary solar events, the Bastille Day event and the Halloween event, have been studied by the characteristic height of the ELF waveguide through measurement of Schumann resonance (SR) parameters (Sátori et al., 2016). The observational results verify the conclusion by Sátori et al. (2005) that the hard solar X-ray has an important role in modifying the Earth-ionosphere cavity, with changes in the electron density in the height range from ~ 90km -100 km.
The aim of the present study is the investigation of the solar flare effects on ionospheric absorption at mid- and low-latitudes taking into account the solar zenith angle with the systematic analysis of the ionospheric fmin parameter measured at different ionosonde stations. Following this introduction, the exact method and the data examined are described in section 2, the results are detailed in section 3. Finally, the results are discussed and the concluding remarks are written in section 4.

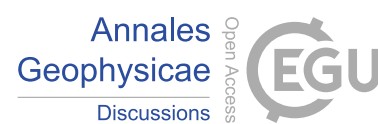

## 2. Method and data

The time series of the fmin parameter have been analyzed during solar flares and solar proton events (SPE) of different intensities during solar cycle 23. The ionospheric parameters have been manually verified before the analysis. The fmin

parameter, representing the lowest recorded ionosonde echo, is usually considered as a qualitative measure of the so called ''nondeviative'' radio wave absorption in the ionosphere (Risbeth and Gariott, 1969). It has been used to investigate the absorption of the D region in the last decades (Lusignan, 1960; Oksman et al., 1981; Kokourov, 2006; Sharma et al, 2010; Schmitter et al, 2011). However, the fmin parameter is dependent on the radar instrumental characteristics and the radio-noise level. In order to minimize and compensate for the instrumental errors, only data measured by Lowell type digisondes (Global

Ionospheric Radio Observatory (GIRO, http://giro.uml.edu) data) have been used during this analysis. Furthermore, a dfmin parameter (difference between the value of the fmin and the mean fmin for reference days) have also been determined for the analysis. At least 10 reference days have been selected before and after the selected flares based on the X-ray radiation ($< 0.5 \ast 10^{-4}$) and proton flux [0.8-4 MeV] ($< 3 \ast 10^3$) measured by GOES satellites. The analysis has been repeated for ionospheric data recorded at meridionally-distributed ionosonde stations (the selected stations with their geographical coordinates are found

in Table 1.).

Three solar events from solar cycle 23. have been selected for analysis when a strong X-class flare has been accompanied by a strong solar proton event. Further conditions that we had to consider in the selection that the European and South African ionosonde stations had to be in the sunlit hemisphere during the flares. Thus, the absorption variation caused by the radiation

could be determined using the fmin parameter measured at these stations. Some less intense (M class) flares which have occurred during the days before and after the X-class flares have also been analyzed. The selected solar flares and accompanying SPEs are listed in Table 2 and 3.

The solar zenith angle of the observation sites has been also taken into account in the case of the selected flare events. We determined the solar zenith angle of the ionospheric stations at the peak time of the selected flare events. Generally, the zenith

angles of the observation sites were large in Europe and small in South Africa in the case of the same flare because there are no GIRO stations between these two regions. Firstly, the solar zenith angle dependence of the duration of the total radio fade-out, and of the first measured value of the fmin and dfmin parameters after the fade-out have been investigated. In the case of the X-class solar flares the radio fade-out took 1-2 hours especially at-stations with low solar zenith angle. Consequently, in the next step we compared the solar zenith angle dependence of the fmin and dfmin parameters detected at the different stations

at a certain time after the fade-out when there were measured data at most of the stations.



## 3. Results

Three time periods (2001-09-23 - 2001-09-28; 2003-10-27 - 2003-11-02; 2006-12-04 - 2006-12-08), when the eight X and M class flares and three SPEs occurred, have been selected for this study. First the fmin, foE, foF2 parameters have been investigated during these special periods (Fig. 1. and Fig. 2.). On the upper plots the variation of the X-rays are shown, while on the second upper plot the changes of the proton flux are shown during the three mentioned special periods. The changes of the fmin, foE and foF2 parameters detected at meridionally distributed stations are seen on the lower panels of the plots from higher to lower latitudes consecutively. The times of the X and M class flares which have been selected for further analysis are indicated by green dashed lines. Total and partial radio fade-out was experienced at every ionospheric station during and after the X class solar flares (on 2001-09-24, 2005-12-05 and on 2003-10-28 (shown later)) and also in the case of some M class flares (e. g. on 2006-12-06). The detected time periods of the total radio fade-out were between 15 min and ~150 min. The observed time of the lack of the reflected echoes is similar to the results of Sahai et al. (2006) detected by ionosondes over the Brazilian sector on 28 October 2003. Extreme increases of the fmin values (4-9 MHz) were observed at almost every stations at the time of the X-class solar flares (on 2001-09-24, 2005-12-05 (Fig. 1. and 2.) and on 2003-10-28 (not shown here). Furthermore, the variation of the fmin parameter was well pronounced (2-7 MHz) during the M class solar flares as well (e. g. on 28 September 2001 and on 06 December 2006, Fig. 1. and 2.). During the time of the increased values of the fmin parameters the co-occurring absence of the foE parameter was detected (Fig. 1. and 2.). There were no detected changes of the fmin parameter at high latitude (Tromso) during the X9.0 (on 5 December, 2006) and M6.0 (on 6. December, 2006) class solar flares. However, total radio fade-out was observed for almost two days at Tromso on 07 and 08 December, 2006 due to the polar cap absorption (PCA) (Fig. 2.) caused by the precipitation of energetic charged particles (Rose and Ziauddin, 1962; Bothmer and Daglis, 2007). Data was not available from high latitude for the periods 2001-09-23 - 2001-09-28 and 2003-10-27 - 2003-11-02. The changes of the dfmin parameter during the selected time periods (2001-09-23 - 2001-09-28; 2003-10-27 - 2003-11-02; 2006-12-04 - 2006-12-08) have been analyzed as well. The variation of the dfmin parameter between 2003-10-27 and 2003-11-02 can be seen on Figure 3. Huge variations also occurred in the value of the dfmin at the time of the X and M class flares on 27 and 28 October, 2003. The detected total radio fade-out, observed at every ionosonde stations at the time of the X17 flare on 28 October 2003, can be seen in Fig. 3. as well. In addition, the observed changes of the dfmin parameter was 4-8 MHz at the time of the X-class flares (e.g. on 28 October, 2003, Fig. 3.) and 1-4 MHz at the time of the M class flares (e. g. 2 flares on 27 October 2003, Fig. 3.).

In the next step of the analysis the solar zenith angle dependence of the duration of the fade-out, the fmin and dfmin parameters have been investigated during and after the time of the selected solar flares. The solar zenith angles of the stations at the time of the peak of the 8 flares have been determined for the analysis. The variation of the X-ray flux and dfmin parameter measured at stations with different solar angles on 27 and 28 October, 2003 are shown here (Fig. 4. and 5.). Looking at the changes of the dfmin parameter at the time of the X17 solar flare on 28 October (Fig. 4.) it seems that the total radio-fadeout and also the measured peak value of dfmin show a solar zenith angle dependence. The duration of the fade-out and also the dfmin go from



smaller to larger values at stations with larger to smaller zenith angle. A similar tendency of the dfmin parameter can be seen during the two M class solar flares on 27 October 2003 (Fig. 5.).

**3.1 Duration of the fade-out**

There were four flares during the selected time periods when total radio fade-out was detected at least at four stations. The solar zenith angle dependence of the duration of the total fade-out has been investigated during these four events. The results can be seen in the Table 4. and in Fig. 6. The solar zenith angle dependence of the duration of the total radio fade-out can be clearly seen on Fig. 6. especially during the X17 flare on 28 October 2003 (Fig. 6a.) and the X9 flare on 5 December, 2006 (Fig. 6b.). The duration of the fade-out tends to increase with decreasing solar zenith angle. The tendency is similar in the other

two cases but is not that pronounced. It has to be mentioned here that generally the number of observations (N) is limited to say anything about statistical significance but the plots are illustrative.

**3.2 Variation of the fmin and dfmin values directly after the fade-out**

   The solar zenith angle dependence of the fmin and dfmin values measured at the peak time of the flares or immediately after

the fade-out has been analyzed in the next step. The results are shown in Table 4 and Fig's. 7. and 8. The solar zenith angle dependence of the fmin and dfmin values can be seen in most cases. The fmin values are increasing with decreasing solar zenith angle. This increasing trend of the fmin parameter is especially pronounced on the plots Fig. 7b., Fig. 7c. and Fig. 7e in the case of the flares 2006-12-05, 2001-09-24 and 2006-12-06 respectively. The trend can be recognized in the plots 7d., 7e, 7g and 7h, although points are more scattered. However, there is no observable trend in Fig. 7a. in the case of the most intense

flare of the Halloween event on 28 October, 2003. Looking at the fmin values during the flares the effect of the different flare intensities on the ionosphere can be detectable as well. The fmin values in the case of the X-class flares in Fig. 7a. (2003-10-28) and in Fig. 7c. (2001-09-24) are larger (fmin > 5 MHz) than in the case of the M class flares from the same periods (3 < fmin < 8 MHz on Fig. 7d., 7f., 7g. and 7h.). A seasonal dependence of the fmin parameter is also evident. The values are larger in September and October (3 < fmin < 11 MHz) than in December (2< fmin < 7 MHz).

The increasing trend with decreasing solar zenith angle is also detectable in the dfmin values. Moreover, the points are not that scattered in the Fig. 8e, 8f, 8g and 8h, in the case of the M class flares. Nevertheless, the increasing trend cannot be seen in Fig. 8a and 8d. during the flares that occurred at 12:43 on 27 and at 11:24 on 28 October, 2003. The explanation for the lack of an increasing trend in these cases can be that the times of the fade-out are very different at the different ionospheric stations (see Fig. 6.). Thus, the first fmin and dfmin values after the fade-out were measured at different times when also the X-ray

radiation of the flare were different. In order to eliminate this possible cause for variability, we analyzed the fmin and dfmin parameters at a certain time after the peak of the flares when there were detectable values at the most stations.





### 3.3 Variation of the fmin and dfmin parameters at a certain time after the fade-out

The results of the comprehensive investigation of the solar zenith angle dependence of the fmin and dfmin values measured at a certain time after the peak of the flares are shown in Table 5 and in Fig. 9 and 10. The exact time when the measurement occurred are shown in the header of different cases in Table 5. and in Fig 9 and 10. The solar zenith angle dependence of the
5   fmin and dfmin values are more conspicuous than in the previous case. The fmin values are increasing with decreasing solar zenith angle in every case, and also after the most intense flare of the Halloween event on 28 October, 2003 (see Fig. 9a.). The solar zenith angle dependence seems well defined in the dfmin values. The increasing trend appears in every case, and also after the flares that occurred at 12:43 on 27 and at 11:24 on 28 October, 2003 (Fig. 10a. and 10d.). Moreover, the points in Fig. 10. are less scattered than in the case of fmin, in Fig. 9.

### 3.4 Comprehensive investigation of the intensity of flares and the solar zenith angle dependence

The results showed that the ionospheric response depended also on the intensity of flare (change in the X-ray flux). The value of the dfmin variation reached 6-9 MHz during and after the X17 (2003-10-28, Fig. 8a.) and X2 (2001-09-24, Fig. 8c.) flares. Whereas the dfmin values varied between 1 and 3 MHz in the cases of the M3.3 and M 2.4 flares on 28 September, 2001 (Fig.
8g. and 8h.). Thus, a comprehensive analysis, taking into account the solar zenith angle and the intensity together, has also been performed. The solar zenith angle and the X-ray radiation dependence of the fmin and dfmin parameters measured at the peak of the flare events or directly after the fade-out are shown in Fig. 11a. and 11b. respectively. The results show that the value of the fmin and dfmin parameters depend on the intensity of the X-ray radiation, but they also depend on the solar zenith angle of the stations where they have been measured. The largest fmin (> 7 MHz) and dfmin (> 5MHz) values have been
detected during the X-class solar flares (X-ray radiation > 2.61E-04 Wm-2) and at the stations with low (< 40 °) solar zenith angle. Since, the exact times of the measurement were different (because of the different duration of the total radio fade-out), this analysis has been repeated for fmin and dfmin values measured at a certain time after the peak of the flares when the parameters were detectable at most of the stations. (The exact observation time and the detected X-ray intensity by GOES satellites at that time are shown in the header of different cases in Table 5.) The results of the analysis are shown in Fig. 12.
The X-ray radiation dependence can be seen in the value of the fmin parameter in this case as well. However, it is much better defined in the case of the dfmin parameter. Larger dfmin values (> 4.5 MHz) are related to the measurements when the X-ray radiation exceeded 3.4E-05 Wm$^{-2}$. Moreover, the lowest fmin and dfmin values were measured when the X-ray radiation was weaker (< 1.33E-05 Wm$^{-2}$) and the solar zenith angle of the stations was above 35 °.

**4. Discussion and conclusion**

The solar flare effects on ionospheric absorption at mid- and low-latitude have been investigated with the systematic analysis of ionosopheric parameters in this study. Three solar events from solar cycle 23. have been selected for analysis when eight X



and M class flares occurred and have been accompanied by three strong solar proton events. The solar zenith angle of the observation sites at the time of the selected flares has also been considered in the analysis.

The lowest recorded ionosonde echo, characterized by the minimum frequency (fmin), has been used as a qualitative measure of the so called ''nondeviative'' radio wave absorption in recent decades (Lusignan, 1960; Oksman et al., 1981; Kokourov, 2006; Sharma et al, 2010; Schmitter et al, 2011). However, a systematic analysis of this parameter measured at different ionospheric stations during solar flares has not been previously investigated. To minimize the instrumental errors a dfmin parameter (the difference between the value of the fmin and the mean fmin for reference days) has also been determined for the analysis.

Extreme increases of the fmin values (4-9 MHz during X class and 2-7 MHz during M class flares) were observed at almost every stations at the time of the flare events. These enhancements of fmin during solar flares are in good agreement with the results reported by Sharma et al. (2010). During the time of the increased values of the fmin parameters the co-occurring absence of the foE parameter was detected. Huge variations (4-8 MHz at the time of the X-class flares and 1-4 MHz at the time of the M class flares) were found in the dfmin parameter as well.

No detected changes were noted in the fmin parameter at high latitude (Tromso) during the X9.0 (on 5 December, 2006) and the M6.0 (on 6 December, 2006) class solar flares. However, total radio fade-out was observed for almost two days at Tromso on 07 and 08 December, 2006 due to the polar cap absorption (PCA) caused by the precipitation of energetic charged particles (Rose and Ziauddin, 1962; Bothmer and Daglis, 2007).

Total and partial radio fade-out were experienced at every ionospheric station during and after the X class solar flares (on 2001-09-24, 2003-10-28, 2005-12-05 and on 2005-12-05) and also in the case of some M class flares (e. g. on 2006-12-06). The observed time of the absence of the echoes were between 15 min and 150 min, similar to the findings of Sahai et al. (2006) with ionosondes over the Brazilian sector on 28 October 2003. Based on the present results, the duration of the total radio fade-out during intense solar flares (> M6) is highly dependent on the solar zenith angle of the observation sites.

The analysis of the fmin and dfmin values measured at the peak time of the flares or right after the fade-out shows a solar zenith angle dependence as well. The fmin and dfmin values are increasing with decreasing solar zenith angle. However, this increasing trend is not clear in the case of the most intense (X2 and X17) solar flares when the duration of fade-out are very different at the different ionospheric stations. Thus, in the next step we analyzed the solar zenith angle dependence of the fmin and dfmin parameters at a certain time after the peak of the flares when there were detectable values at most stations. The solar zenith angle dependence of the fmin and dfmin parameters is more conspicuous than in the previous case. The fmin and dfmin





values are increasing with decreasing solar zenith angle in every case. Moreover, they are less scattered. However, Li et al. (2018) concluded that there is no strong relationship between the Ne variation of the D region and the solar zenith angle.

According to the results of Liu et al (2018) there is a large correlation between the flare-induced Ne enhancement in the D-layer and the X-ray flux intensity of the flare. In order to study the impact of the X-ray flux on the fmin and dfmin parameters a comprehensive analysis, taking into account the solar zenith angle and the intensity of the flare together, has also been performed. The results show that the values of the fmin and dfmin parameters are highly dependent on the X-ray radiation intensity, but they also depend on the solar zenith angle of the stations where they have been measured.

Based on the results, the dfmin parameter is a good qualitative measure for the relative variation of the "non-deviative" absorption especially in the case of the less intense solar flares which do not cause total radio fade-out in the ionosphere (class < M6). These measurements may inform models in the future in describing the changes in ionospheric absorption during solar flares with different intensities. However, further analysis of this ionosonde parameter and its comparison with other techniques to measure the ionospheric absorption are necessary to confirm its use as a reliable index.

## 5. Acknowledgement

The contribution of V. Barta was supported by the United States Department of State Bureau of Educational and Cultural Affairs as part of a Fulbright Visiting Scholar Program to the Massachusetts Institute of Technology, Cambridge, USA and by the GINOP-2.3.2-15-2016-00003 project. The contribution of G. Sátori was supported by the National Research, Development and Innovation Office, Hungary-NKFIH, K115836. The authors are grateful to the University of Massachusetts Lowell Center for Atmospheric Research for the Digisonde data and SAO-X program for data processing. Data from the South African Ionosonde network is made available through the South African National Space Agency (SANSA), who are acknowledged for facilitating and coordinating the continued availability of data. This paper uses data from the Juliusruh Ionosonde which is owned by the Leibniz Institute of Atmospheric Physics Kuehlungsborn. The responsible Operations Manager is Jens Mielich. This paper uses ionospheric data from the USAF NEXION Digisonde network, the NEXION Program Manager is Mark Leahy. The authors wish to thank the OMNIWeb data center for providing Web-accessible to the solar data of the Geostationary Operational Environmental Satellites (GOES) satellites.



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



| Ionospheric Station | Latitude (°) | Longitude (°) |
|---|---|---|
| Tromso | 69.6 | 19.2 |
| Juliusruh | 54.6 | 13.4 |
| Chilton | 51.5 | 359.4 |
| Pruhonice | 50 | 14.6 |
| Rome | 41.9 | 12.5 |
| San Vito | 40.6 | 17.8 |
| Ascension Isl. | -7.95 | 345.6 |
| Madimbo | -22.39 | 30.88 |
| Grahamstown | -33.3 | 26.5 |

**Table 1.** The selected ionosonde stations and their geographical coordinates.

| Selected time period | | | | |
|---|---|---|---|---|
| Date | X-ray class | Start | Peak | End |
| 2001-09-23 - 2001-09-28 | | | | |
| 2001-09-24 | X2.6 | 9:32 | 10:38 | 11:09 |
| 2001-09-28 | M3.3 | 8:10 | 8:30 | 9:10 |
| 2001-09-28 | M2.4 | 9:34 | 10:14 | 10:50 |
| 2003-10-27 - 2003-11-02 | | | | |
| 2003-10-27 | M5 | 9:21 | 9:27 | 9:32 |
| 2003-10-27 | M6.7 | 12:27 | 12:43 | 12:52 |
| 2003-10-28 | X17 | 09:51 | 11:10 | 11:24 |
| 2006-12-04 - 2006-12-08 | | | | |
| 2006-12-05 | X9.0 | 10:18 | 10:35 | 10:45 |
| 2006-12-06 | M6.0 | 8:02 | 8:23 | 9:03 |

**Table 2**. List of selected flare events for this study.

| Start Date | Maximum | Proton Flux (pfu @ >10 MeV) | Associated flare |
|---|---|---|---|
| 2001-09-24 12:15 [UTC] | 2001-09-25 22:35 [UTC] | 12900 | 2001-09-24 10:38 [UTC] |
| 2003-10-28 12:15 [UTC] | 2003-10-29 06:25 [UTC] | 29500 | 2003-10-18 11:10 [UTC] |
| 2006-12-06 15:55 [UTC] | 2006-12-07 19:30 [UTC] | 1980 | 2006-12-05 10:35 [UTC] |

5  **Table 3.** List of the SPEs (and their proton fluxes) which followed the X-class solar flares.



| X-ray class and time of the solar flare [UTC] | | | | | X-ray class and time of the solar flare [UTC] | | | | |
|---|---|---|---|---|---|---|---|---|---|
| Station name | Solar zenith angle | Duration of fade-out | fmin | dfmin | Station name | Solar zenith angle | Duration of fade-out | fmin | dfmin |
| X17, 2003-10-28, 11:24 | | | | | M6.0, 2006-12-06, 08:23 | | | | |
| Juliusruh | 67.77 | 15 | 8.45 | 6.7 | Pruhonice | 79 | 0 | 2.5 | 0.5 |
| Chilton | 65.15 | 50 | 10.35 | 7.8 | Rome | 73.19 | 15 | 4.6 | 3.1 |
| Rome | 55.07 | 75 | 8.5 | 5.8 | San Vito | 69.98 | 0 | 3.6 | 1.9 |
| San Vito | 54.06 | 30 | 7.4 | 4.75 | Ascension Isl. | 63.51 | 75 | 5.6 | 3.2 |
| Grahamstown | 26.09 | 150 | 6.7 | 4.2 | Grahamstown | 23.49 | 75 | 6.4 | 3.6 |
| Ascension Isl. | 22.9 | 135 | 10.1 | 7 | Madimbo | 18.16 | 90 | 6.9 | 4.1 |
| X9, 2006-12-05, 10:35:00 | | | | | M5.0, 2003-10-27, 09:27 | | | | |
| Pruhonice | 72.5 | 30 | 4.15 | 2.146 | Juliusruh | 69.56 | 0 | 4.25 | 3.1 |
| Rome | 64.64 | 60 | 4.6 | 2.06 | Chilton | 71.22 | 0 | 3.95 | 2.3 |
| San Vito | 63.05 | 30 | 4.2 | 2.42 | Rome | 58.13 | 0 | 6.6 | 3.2 |
| Ascension Isl. | 36.14 | 60 | 6.6 | 4.1 | San Vito | 55.41 | 0 | 5.1 | 3.3 |
| Grahamstown | 12.29 | 75 | 6.7 | 3.6 | Grahamstown | 21.77 | 150 | 6.15 | 2.4 |
| Madimbo | 9.63 | 90 | 6.1 | 3.5 | Ascension Isl. | 47.96 | 0 | 7.5 | 4.8 |
| X2, 2001-09-24, 10:38 | | | | | M3.3, 2001-09-28, 08:30 | | | | |
| Juliusruh | 55.31 | 45 | 5.3 | 3.8 | Chilton | 68.97 | | 3.65 | 1.7 |
| Chilton | 54.62 | 30 | 6.2 | 3.9 | Juliusruh | 64.28 | | 3.66 | 1.8 |
| Rome | 42.76 | 180 | 7.05 | 2.9 | Rome | 55.63 | | 7.45 | 2.7 |
| Grahamstown | 33.69 | 0 | 10.6 | 8.1 | Grahamstown | 38.25 | | 5.35 | 3.05 |
| Madimbo | 25.05 | 90 | 7.95 | 5.6 | Madimbo | 27.86 | | 6.55 | 3.1 |
| M6.7, 2003-10-27, 12:43 | | | | | M2.4, 2001-09-28, 10:14 | | | | |
| Juliusruh | 71.41 | 0 | 4.25 | 3.2 | Chilton | 57.85 | | 3.2 | 0.9 |
| Chilton | 65.34 | 0 | 4.9 | 2.8 | Juliusruh | 57.42 | | 3.86 | 2.1 |
| Rome | 60.11 | 0 | 7.6 | 4 | Rome | 45.27 | | 8.95 | 2.6 |
| San Vito | 61.34 | 0 | 5.1 | 3.4 | Grahamstown | 31.25 | | 4.8 | 2.35 |
| Grahamstown | 42.8 | 0 | 4.85 | 1.9 | Madimbo | 21.25 | | 6.5 | 2.9 |
| Ascension Isl. | 4.82 | 15 | 6.7 | 3.1 | | | | | |

**Table 4.** The ionosonde stations (first column) with their solar zenith angle (second column) at the time of the peak of the selected solar flares. The duration of the total radio fade-out at the station appear in the third column. The tabulated fmin (4th column) and dfmin (5th column) values were measured at the peak time of the flares or directly after the fade-out.





| Intensity of X-ray rad. [Wm$^{-2}$], time of the measurement [UTC] | | | | | Intensity of X-ray rad. [Wm$^{-2}$], time of the measurement [UTC] | | | | |
|---|---|---|---|---|---|---|---|---|---|
| Station name | Solar zenith angle | Duration of fade-out | fmin | dfmin | Station name | Solar zenith angle | Duration of fade-out | fmin | dfmin |
| 7.89E-05, 2003-10-28, 13:30 | | | | | 8.22E-06, 2006-12-06, 10:00 | | | | |
| Juliusruh | 74.68 | 15 | 2.9 | 1.9 | Pruhonice | 73.39 | 0 | 2.05 | 0 |
| Chilton | 67.64 | 50 | 5.5 | 3.7 | Rome | 65.89 | 15 | 2.35 | 0 |
| Rome | 64.41 | 75 | 6.3 | 1.9 | San Vito | 63.76 | 0 | 2.1 | 0.4 |
| San Vito | 66.26 | 45 | 3.8 | 2.6 | Ascension Isl. | 42.95 | 75 | 5.6 | 3.2 |
| Grahamstown | 50.09 | 150 | 6.7 | 3.6 | Grahamstown | 10.86 | 75 | 4.8 | 2.1 |
| Ascension Isl. | 10.81 | 135 | 10.1 | 6.6 | Madimbo | 2.9 | 90 | 6.9 | 4.1 |
| 1.33E-05, 2006-12-05, 12:00 | | | | | 3.39E-05, 2003-10-27, 09:30 | | | | |
| Pruhonice | 73.87 | 30 | 2 | 0.1 | Juliusruh | 69.56 | 0 | 4.25 | 2.9 |
| Rome | 65.7 | 60 | 4.4 | 1.7 | Chilton | 71.22 | 0 | 3.95 | 1.8 |
| San Vito | 65.68 | 30 | 2.1 | 0.4 | Rome | 58.13 | 0 | 6.6 | 3.2 |
| Ascension Isl. | 18.53 | 60 | 6.6 | 4 | San Vito | 55.41 | 0 | 5.1 | 3.3 |
| Grahamstown | 27.65 | 75 | 6.7 | 4 | Ascension Isl. | 47.96 | 0 | 7.5 | 4.8 |
| Madimbo | 30.66 | 90 | 6.1 | 3.5 | Grahamstown | 21.77 | 150 | Nan | Nan |
| 6.06E-05, 2001-09-24, 11:30 | | | | | 3.26E-05, 2001-09-28, 08:30 | | | | |
| Juliusruh | 55.57 | 45 | 4.35 | 3 | Chilton | 68.97 | 0 | 3.65 | 1.7 |
| Chilton | 52.34 | 30 | 5.85 | 3.4 | Juliusruh | 64.28 | 0 | 3.66 | 1.8 |
| Rome | 42.94 | 180 | Nan | Nan | Rome | 55.63 | 0 | 7.45 | 2.7 |
| Grahamstown | 38.2 | 0 | 5.9 | 3.5 | Grahamstown | 38.25 | 0 | 5.35 | 3.05 |
| Madimbo | 32.88 | 90 | 7.95 | 5.6 | Madimbo | 27.86 | 0 | 6.55 | 3.1 |
| 1.59E-05, 2003-10-27, 13:00 | | | | | 2.53E-05, 2001-09-28, 10:15 | | | | |
| Juliusruh | 72.68 | 0 | 4.25 | 1.2 | Chilton | 57.85 | 0 | 3.2 | 0.9 |
| Chilton | 66.18 | 0 | 4.9 | 1.8 | Juliusruh | 57.42 | 0 | 3.86 | 2.1 |
| Rome | 61.83 | 0 | 7.6 | 0.85 | Rome | 45.27 | 0 | 8.95 | 2.6 |
| San Vito | 63.38 | 0 | 5.1 | 2.1 | Grahamstown | 31.25 | 0 | 4.8 | 2.35 |
| Grahamstown | 46.19 | 0 | 4.85 | 1.9 | Madimbo | 21.25 | 0 | 6.5 | 2.9 |
| Ascension Isl. | 6.64 | 15 | 6.7 | 3.1 | | | | | |

**Table 5.** The value of the X-ray radiation in Wm$^{-2}$ and the date and exact time when the measurement occurred are shown in the header in every case. The ionosonde stations (first column) with their solar zenith angle (second column) at the time of the measurement after the peak of the flares. The duration of the total radio fade-out at the station appear in the third column. Also included are the measured fmin (4th column) and dfmin (5th column) values at the time of the measurement after the peak of the flares.





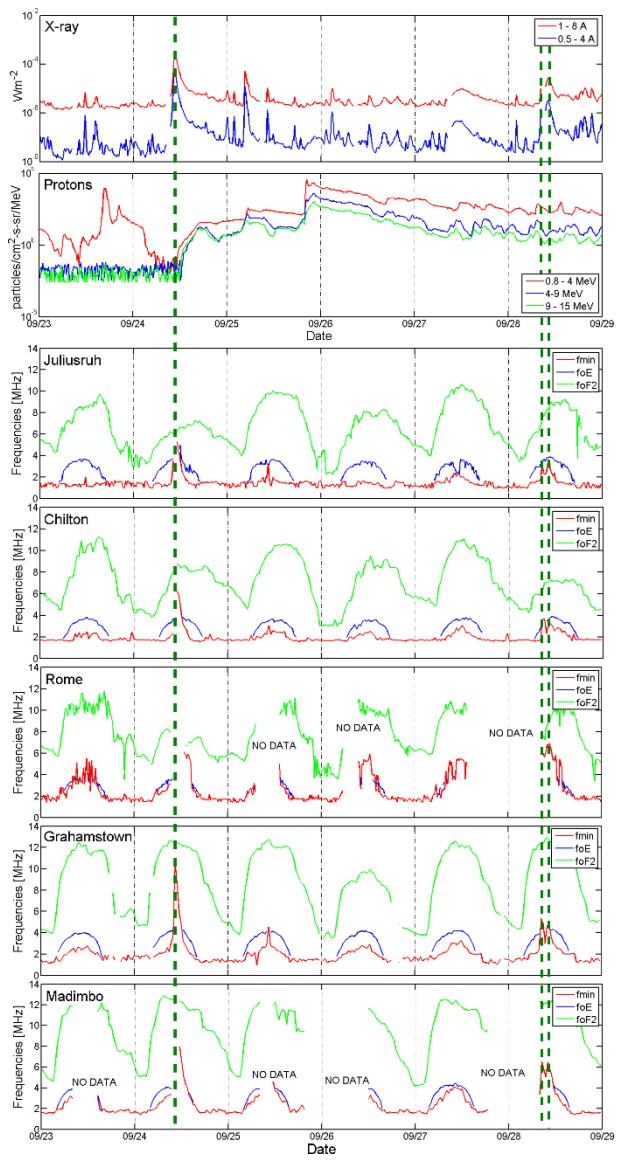

**Figure 1.** The variation of the X-ray flux (upper plot), proton flux (second upper plot), the changes of the fmin (red lines), the foE (blue lines) and the foF2 (green lines) parameters detected at different ionospheric stations from higher to lower latitudes consecutively between 2001-09-23 and 2009-09-28. The vertical green dashed lines show the time of the selected flares.





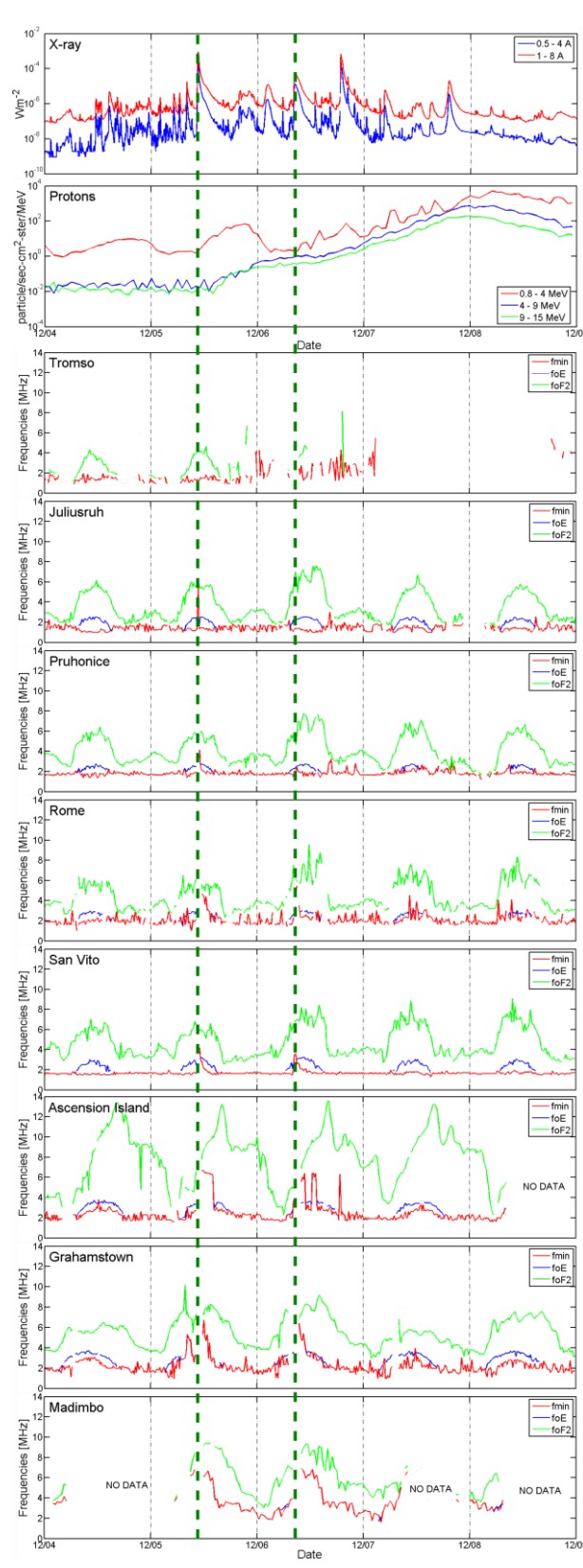

**Figure 2.** The variation of the X-ray (upper plot) flux, proton flux (second upper plot), and the changes of the fmin (red lines), the foE (blue lines) and the foF2 (green lines) parameters detected at different ionospheric stations from higher to lower latitudes consecutively between 2006-12-04 and 2006-12-08. The vertical green dashed lines show the time of the selected flares.





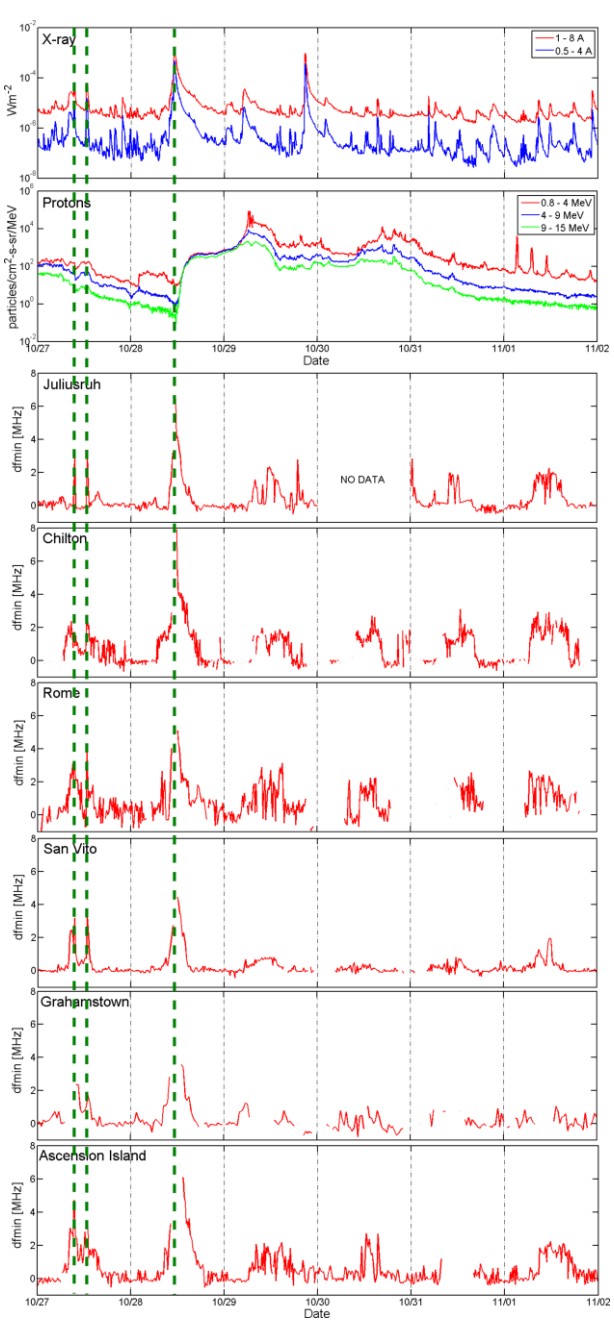

**Figure 3.** The variation of the X-ray flux (upper plot), proton flux (second upper plot), and the changes of the dfmin (red lines) parameter detected at different ionosonde stations from higher to lower latitudes consecutively between 2003-10-27 and 2003-11-01. The vertical green dashed lines show the time of the selected flares.





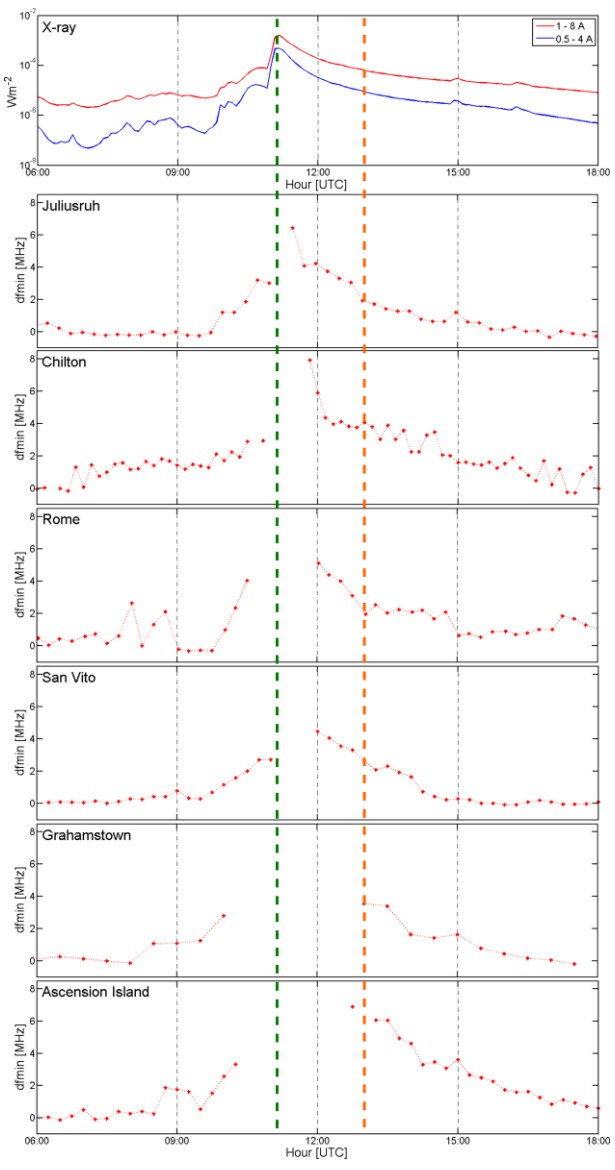

**Figure 4.** The variation of the X-ray flux (upper plot), the changes of the dfmin (red dots and red dashed line) parameter detected at different ionosonde stations with different zenith angle (from larger to smaller) on 28 October 2003 between 06:00 and 18:00 UTC. The vertical green dashed line shows the peak time of the X17 flare while the vertical orange dashed line shows the time used for the second comparison (in Sec. 3.3).



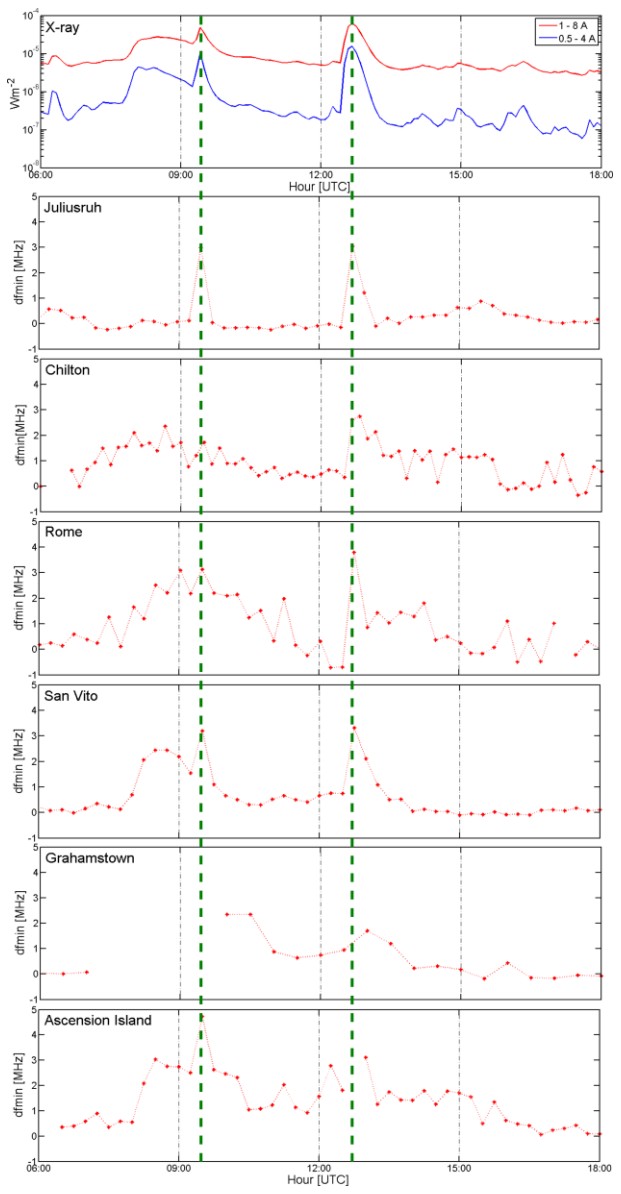

**Figure 5.** The variation of the X-ray (upper plot) and the changes of the dfmin (red dots and red dashed line) parameter detected at different ionospheric stations with different zenith angle (from larger to smaller) on 27, October 2003 between 06:00 and 18:00 UTC. The vertical green dashed lines show the time of the M5 (peak at 9:27 UTC) and M6.7 (peak at 12:43 UTC) flares.



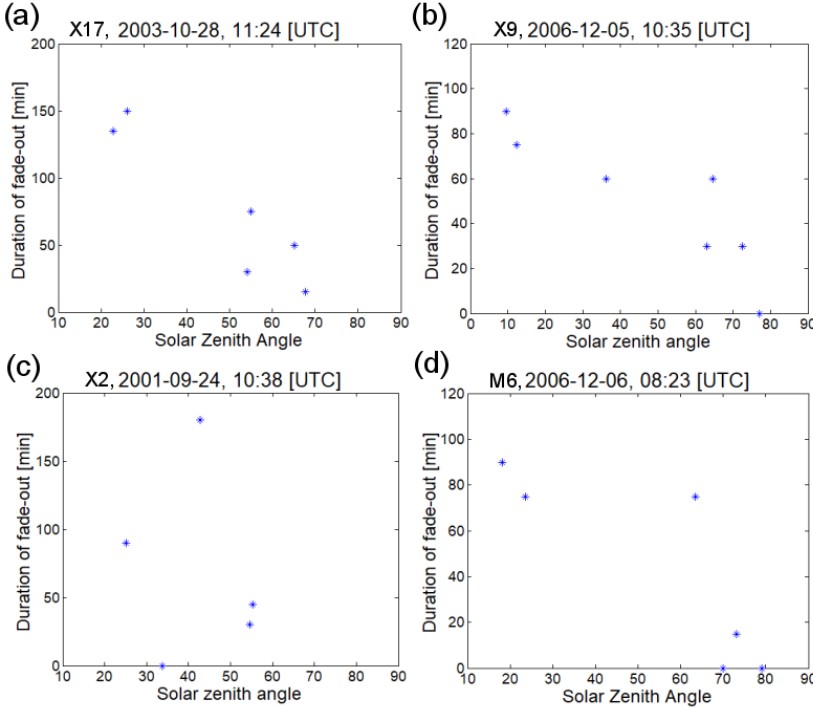

**Figure 6.** The solar zenith angle of the ionosonde stations at the time of the peak versus the measured duration of the total radio fade-out in the case of flare events which occurred on 28 October 2003 (a), on 5 December 2006 (b), on 24 September 2001 (c), and on 6 December 2006 (d).



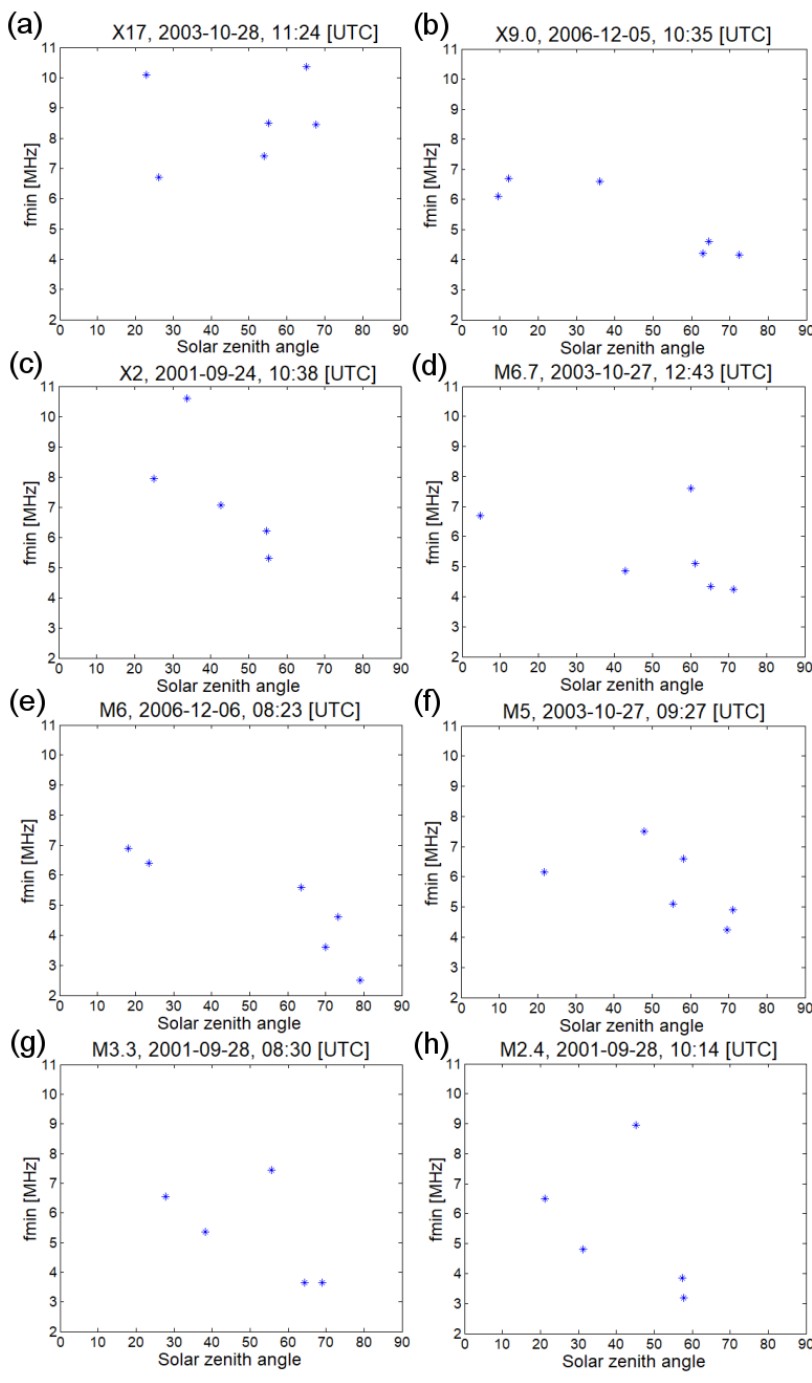

**Figure 7.** The solar zenith angle of the ionosonde stations at the time of the peak versus the fmin value at the peak of the flare events or after the fade-out. The X-ray class and peak time of the solar flares are seen in the title of the different plots. The results related to different flares from high to lower intensities are shown from a to h plots, respectively.




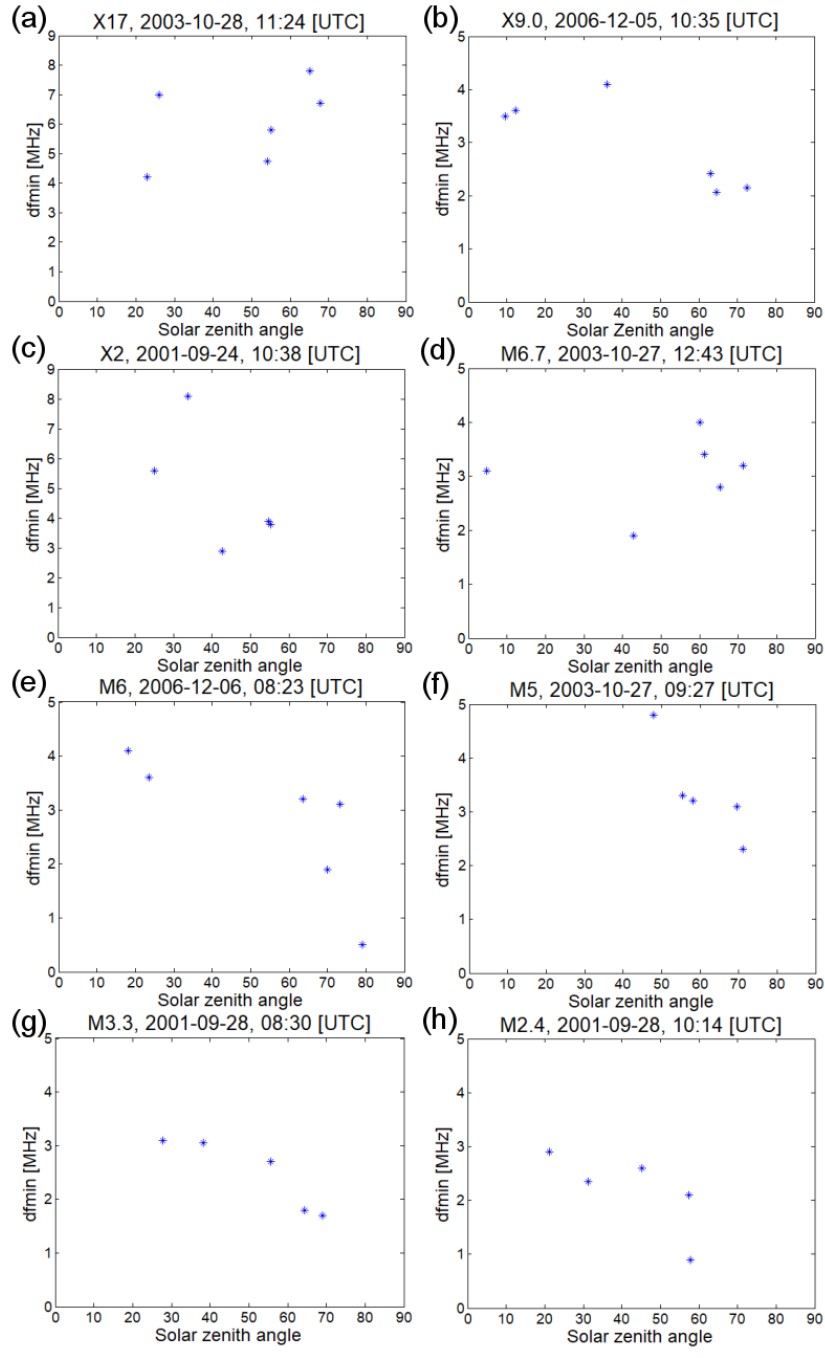

**Figure 8.** The solar zenith angle of the ionosonde stations at the time of the peak versus the dfmin value at the peak of the flare events or after the fade-out. The X-ray class and peak time of the solar flares are seen in the title of the different plots. The results related to different flares from high to lower intensities are shown from a to h plots, respectively.





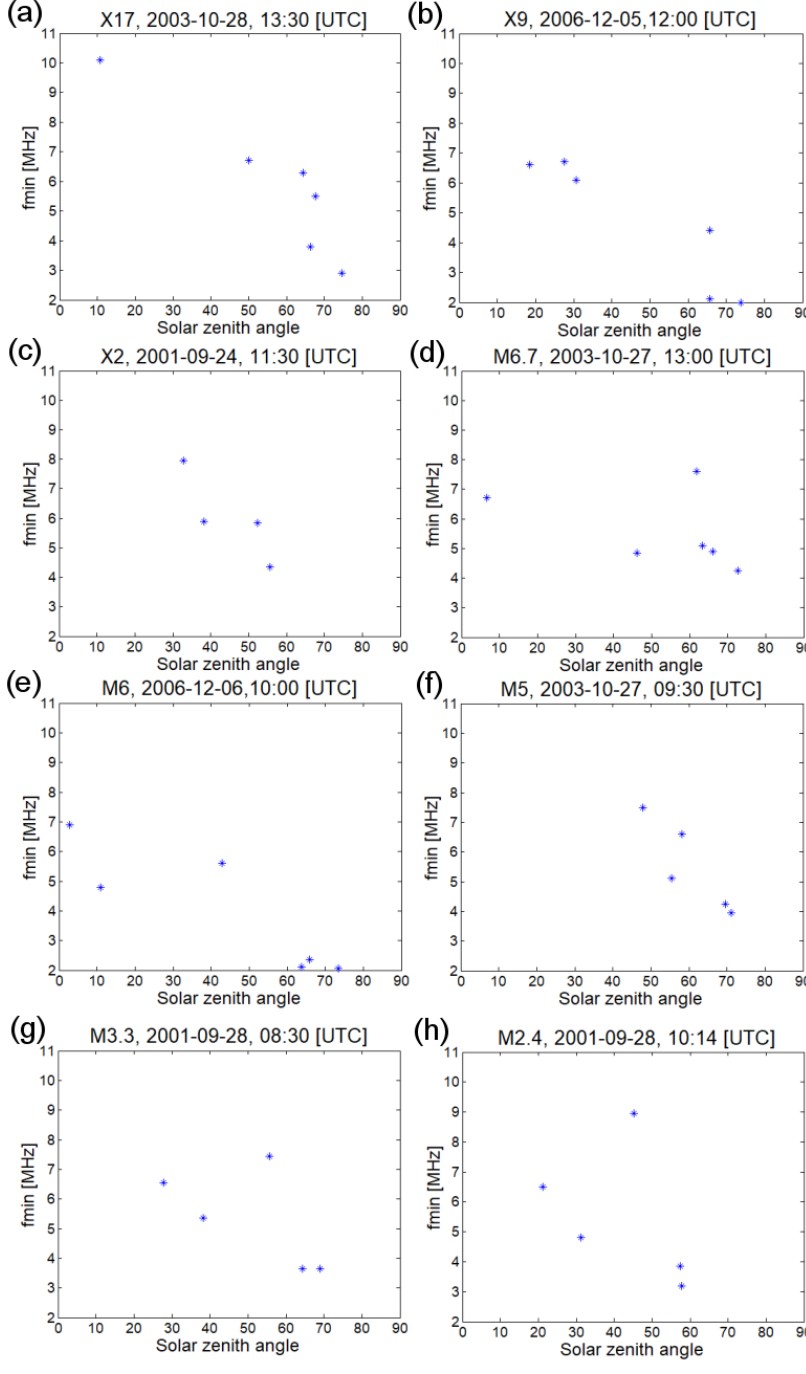

**Figure 9.** The solar zenith angle of the ionosonde stations at a certain time after the peak of the flares versus the fmin value at that time. The X-ray class of the flares and the time when the measurement occurred are shown in the title of the different plots. The results related to different flares from high to lower intensities are shown from a to h plots, respectively.

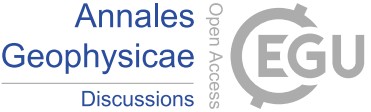



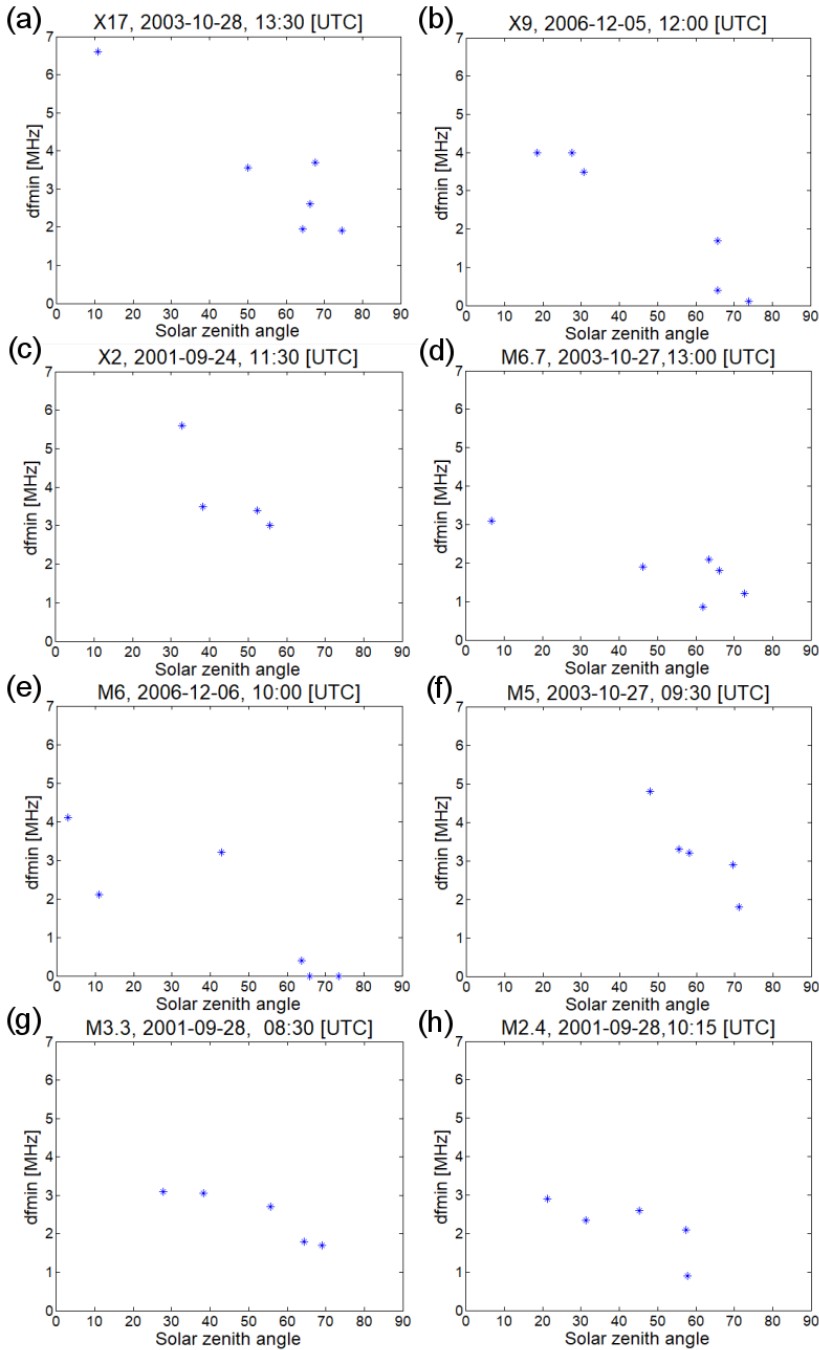

**Figure 10.** The solar zenith angle of the ionosonde stations at a certain time after the peak of the flares versus the dfmin value at that time. The X-ray class of the flares and the time when the measurement occurred are shown in the title of the different plots. The results related to different flares from high to lower intensities are shown from a to h plots, respectively.



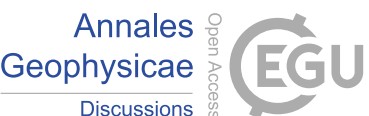

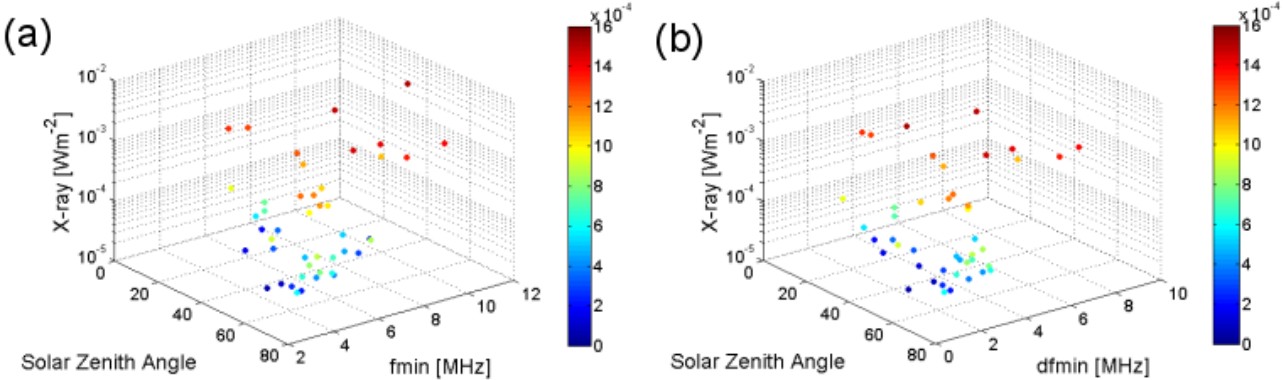

**Figure 11.** The solar zenith angle of the ionosonde stations at the time of the peak, the X-ray radiation at the peak and the value of the fmin (a) and dfmin (b) parameters at the peak of the flare events or after the fade-out. In order to represent the X-ray radiation dependence a colorbar has been connected to the different measurements during the flares with different intensities. The colorbar shows the X-ray radiation in $Wm^{-2}$.

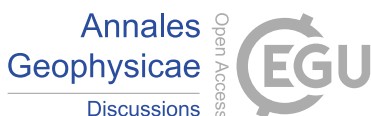

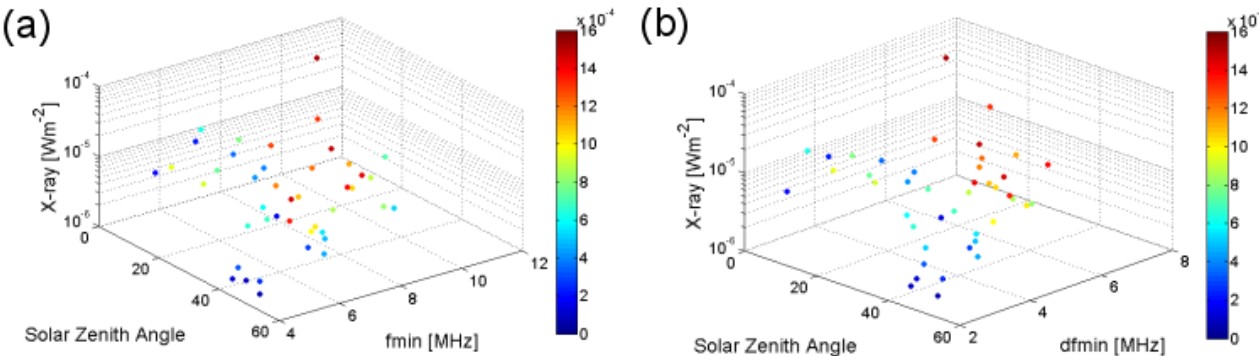

**Figure 12.** The solar zenith angle of the ionosonde stations at the measurement time, the X-ray radiation at the measurement time and the value of the fmin (a) and dfmin (b) parameters measured at a certain time after the peak of the flares (see text). In order to represent the X-ray radiation dependence a colorbar has been assigned to the different measurements as in Figure 11 in the previous case. The colorbar shows the X-ray radiation in Wm-2.

