# Peer review of "Effects of solar flares on the ionosphere as shown by the dynamics of ionograms recorded in Europe and South Africa"

_Annales Geophysicae, 2019_

## Referee Comment (RC1) · Anonymous Referee #1 · 19 Feb 2019

Comments on manuscript "Investigation of the ionospheric absorption response to flare events during the solar cycle 23 as seen by European and South African ionosondes"

The analysis of the absorption induced by the solar flares was performed in this paper. The ionosonde data located at different latitudes were considered. The methods of the fmin and dfmin were applied.

I think this is interesting paper. The problem of the absorption due to solar events is one of the topical problem in the wave propagation investigation. In the paper a lot of

data were analysed.

Main concern: In section Results: "These measurements may inform models in the future in describing the changes in ionospheric absorption during solar flares with different intensities." What models do you mean? How results of this paper could be used in them? For example, the main outcome of the D-RAP models [https://www.swpc.noaa.gov/products/d-region-absorption-predictions-d-rap/] is a global map of the absorption, corresponding to a number of operating frequencies.

I suggest the following minor revision before it is published:

1. I suppose that the structure of the paper is being difficult to understand. 1) The section Introduction is very long. I suggest reducing the part regarding proton precipitation in polar cap (really it is not the main topic of the paper). 2) I suggest to split "Introduction" into a few of paragraphs, otherwise the perception of the text is obstructed. 3) Please, in section "Data and Methods" separate one item from another. 4) The understanding of section "Results" seems to be difficult. I thing that the structure of this section should be change. This section would be better for understanding corresponding to following scheme: particular issue of research, the figures description, provisional conclusion.

2. The sense of the paragraphs (Page 2-3 (30)) in "Introduction" is not clear to me. "The electron density (Ne) of the D region is enhanced by up to one order of magnitude down to about 55 km prior to, during and after the solar proton event (SPE) on January 17, 2005. The largest Ne are found during the maximum of the X-ray flare on January 17. The electron density is still enhanced on January 18 when the X-ray flare decayed but the solar proton fluxes are still enhanced (Singer et al., 2011)." Is it continuation of the paragraph about Patterson et al. (2001) or not?

3. Page .2 (10) I suggest to replace citing Sauer and Wilkinson (2008) text (in italics) by your expression.

4. Please clarify the reason of the analysis of the critical frequencies foE and foF2 (Fig. 1,2) while the absorption is the main subject of the paper.

5. As I understand "The aim of the present study is the investigation of the solar flare effects on ionospheric absorption at mid- and low-latitudes taking into account the solar zenith angle....". I would like once again to draw you attention to the D-RAP model (which is definitely based on the solar zenith angle dependence). I think that this model should be mention in your paper. The correlation between your results and D-RAP model results should be discussed.

6. I suppose that the number of references to other authors in section "Results" could be diminished.

7. Table 2. Please, mention in the caption the unit ("UT", probably).

8. Table 3 and 4. Please, add the units in the titles "Solar zenith angle", "Duration of fade-out", "fmin" and "dfmin".

9. I suppose that the values of the fmin have to be presented with equal (and reasonable) precision (perhaps one decimal point) in the Tables 3 and 4. The similar issue with dfmin.

10. Figure 1. I thing it is better to write "first panel", "second panel" etc., instead of "upper plot" and "second upper plot"

11. It seems to me that axes labels size and titles font size in the Figures 2-5 looks like very small.

---

## Referee Comment (RC2) · Anonymous Referee #2 · 9 Mar 2019

Comments on the manuscript Angeo-2019-14

'Investigation of the ionospheric absorption response to flare events during the solar cycle 23 as seen by European and South African ionosondes'

Submitted to Annales Geophysicae By Veronika Bartaet al.

General Comments: This work shows an analysis of ionospheric parameters in mid- and low-latitudes in relation to solar flares occurred in solar cycle 23. The authors investigated the radio wave absorption in D layer, in which they defined a dfmin parameter as a good qualitative measurement to analyze this absorption. They show an interest analysis with interesting results. However, the authors needs to organize the results and deepen in the physical discussions. Therefore, the authors need to improve significant modifications. This paper needs a major revision. Furhermore, the authors need to improve English significantly.

Major Comments: 1. Abstract: The abstract is not well written. I do not understand the main objective of this study. There are some typo English mistakes as "mimimum", "ionosopheric". The authors need to clarify better the purpose of this work. 2. Introduction (pag. 2, line 25): The solar flares cause an extra ionization in the D region, which causes an absorption of the HF waves, impairing the visualization of the E region in the data (ionograms, for example), and partially or totally in the F region. The authors affirm that there is an absorption in the E region, also. Please, clarify this part. 3. Introduction (pag. 3, line 32): It is necessary to define the fmim parameter; fmim of the F region, E region or both regions? The definition in the section "Method and data" is not enough to understand this part. The authors mention only the discussions about the fmim to be the minimum frequency of ionosphere, but in results (form of the data), I believe that fim refers to the F region. Please, clarify this part. 4. Results: The results are interesting. Although this absorption is well known in the ionospheric data (Denardini et al, 2017, doi: 10,116/s40623-016-0456-7, Sahai et al., cited by authors, and other authors), the relation with the solar zenith angle is present in different form. However, the results are arranged in numerous figures and presented with a confusing text. It would be better to present the figures together (for example Figures 1 and 2 are a single figure)? 5. Discussions and conclusions: The part of the discussion is actually a conclusion. The authors did not elaborate on the physical discussions. There are numerous studies about the subject of relation between flare solar and ionospheric parameters. I suggest that the authors to discuss further the results, that are very interesting, before being published in this journal.

Minor Comments: • English needs to improve in all manuscript: grammar, typo

mistakes, absence of commas, and verbal agreement. • Legend of the figures (1 up to 5) are very difficult to see.

Please also note the supplement to this comment: https://www.ann-geophys-discuss.net/angeo-2019-14/angeo-2019-14-RC2-supplement.pdf

---

## Author Comment (AC1) · 18 Apr 2019

Comments on manuscript "Investigation of the ionospheric absorption response to flare events during the solar cycle 23 as seen by European and South African ionosondes"

The analysis of the absorption induced by the solar flares was performed in this paper. The ionosonde data located at different latitudes were considered. The methods of the fmin and dfmin were applied.

I think this is interesting paper. The problem of the absorption due to solar events is

one of the topical problem in the wave propagation investigation. In the paper a lot of data were analysed.

We thank the Reviewer 1 for expressing their appreciation on our work. We addressed all their points by providing changes/additions in the revised manuscript, and responses here to their comments. We believe our paper is now improved by making changes and short additions throughout the text, mostly in the introduction and discussion. A few more references have also been added.

We hope that the revised paper will now meet with the referee's approval. The changes in the manuscript which have been performed based on the first referee's questions/comments are indicated by pink.

Main concern: In section Results: "These measurements may inform models in the future in describing the changes in ionospheric absorption during solar flares with different intensities." What models do you mean? How results of this paper could be used in them? For example, the main outcome of the D-RAP models [https://www.swpc.noaa.gov/products/d-region-absorption-predictions-d-rap/] is a global map of the absorption, corresponding to a number of operating frequencies.

Thank you for the referee to call our attention to the D-RAP model. We have already known this model and we agree that it is important to mention it in our manuscript. We completed the introduction part of the manuscript with a paragraph about this model:

". . . describing, modelling and monitoring of the ionospheric absorption is an important issue from a practical point of view as well. The process of the ionospheric absorption has been described more extensively by Davies (1990) and Sauer and Wilkinson (2008). Based on these studies the Space Weather Prediction Center (SWPC) has developed a model (D-Region Absorption Prediction, D-RAP2, https://www.swpc.noaa.gov/products/d-region-absorption-predictions-d-rap ) to predict the ionospheric absorption in the D-region. The product provides graphical information about High Frequency (HF) radio propagation conditions around the globe. According

to the model the Highest Affected Frequency (HAF) is largest at the sub-solar point and it decreases with increasing solar zenith angle, $\chi$ (the frequencies taper off from the maximum as $(\cos\chi)^{0.75}$)."

What models do you mean? How results of this paper could be used in them?

Unfortunately, we can not compare the D-RAP model with our results quantitatively because the absorption can not directly be calculated from the fmin parameter. Furthermore, the D-RAP data is available since 2012 (https://www.ngdc.noaa.gov/stp/drap/data/) and we analyzed flare events which occurred between 2001 and 2006 during the solar cycle 23. However our systematic analysis of ionograms can give information on the frequencies below them the sounding electromagnetic waves suffer complete attenuation up to the height of the first reflection (fmin) in the ionosphere depending on the flare intensity (X-ray) and solar zenith angle.

Moreover, these results can contribute to refine the existing (D-RAP model) and future models to describe the changes in ionospheric absorption during solar flares with different intensities. The performance of the D-RAP model has been evaluated with respect to observations at several riometer stations during a representative set of historic events (https://www.ngdc.noaa.gov/stp/drap/DRAP-V-Report1.pdf). However, riometers operate only at high latitudes and at high frequencies. The data investigated in the report have been measured at station with higher latitude than $50°$ and at f >= 30 MHz. Further studies based on our results using the systematic analysis of fmin can help to refine the model describe the attenuation in response to solar flares (described by D-RAP as well) at lower operating frequencies (2-10 MHz) at mid-, and low-latitudes.

Nevertheless, we deleted this general sentence from the discussion part: "These measurements may inform models in the future in describing the changes in ionospheric absorption during solar flares with different intensities." Instead we added the following to the text: "Therefore, our observations confirm the results of Zhang and Xiao
(2005), Spirathi et al. (2013) and the D-RAP model that the solar zenith angle plays an important role in the ionospheric response to solar flares."

I suggest the following minor revision before it is published:

1. I suppose that the structure of the paper is being difficult to understand.

1) The section Introduction is very long. I suggest reducing the part regarding proton precipitation in polar cap (really it is not the main topic of the paper).

Thank you for the suggestion. The paragraph related to the Polar Cap Absorption, and another paragraph about the VLF and ELF records have been deleted from the Introduction. Furthermore, we deleted all the sentences, tables etc. in connection with the Solar Proton Event and PCA. We agree that it is not the main topic of the paper.

2) I suggest to split "Introduction" into a few of paragraphs, otherwise the perception of the text is obstructed.

The introduction has been splited into a few paragraphs.

3) Please, in section "Data and Methods" separate one item from another.

We separated the Method and the Data parts into different paragraphs. Moreover, we completed the data part with the source of the used data.

4) The understanding of section "Results" seems to be difficult. I thing that the structure of this section should be change. This section would be better for understanding corresponding to following scheme: particular issue of research, the figures description, provisional conclusion.

Thank you for your suggestions, we changed the structure of the results part based on them. In the first paragraph we determined the particular issue of research: "In the present study we investigated the response of ionospheric absorption to solar flares with particular interest of the solar zenith angle dependence variation of it. We used ionograms measured at ionosonde stations under different solar zenith angle for the

analysis. We calculated the solar zenith angles of the stations at the time of the peak of the 8 flares for the analysis. We examined three parameters that can be determined from ionograms: duration of the total radio fade-out, the value of the fmin parameter and the value of the dfmin parameter. In the first step we analyzed how the duration of the fade-out during the flare event depended on the solar zenith angle (Sec. 3.1). Secondly the solar zenith angle dependence of the fmin and dfmin parameters measured just after the fade-out were investigated (Sec 3.2). Then we repeated the analysis for the fmin and dfmin parameters measured at a certain time after the fade-out when we again recorded them at all the stations (3.3). In the last step the impact of the intensity variation on the absorption has been considered (3.4)." Then we wrote the figures descriptions with some provisional conclusions. 2. The sense of the paragraphs (Page 2-3 (30)) in "Introduction" is not clear to me. "The electron density (Ne) of the D region is enhanced by up to one order of magnitude down to about 55 km prior to, during and after the solar proton event (SPE) on January 17, 2005. The largest Ne are found during the maximum of the X-ray flare on January 17. The electron density is still enhanced on January 18 when the X-ray flare decayed but the solar proton fluxes are still enhanced (Singer et al., 2011)." Is it continuation of the paragraph about Patterson et al. (2001) or not?

Not, this sentence is related to the study of Singer et al., 2011. However, this part of the introduction has been deleted from the manuscript based on you previous suggestion.

3. Page .2 (10) I suggest to replace citing Sauer and Wilkinson (2008) text (in italics) by your expression.

The part indicated by italics has been replaced by the following text: "The physical background of the ionospheric radio wave absorption mechanism is that the electrons accelerated by the electric field of the propagating radio waves collide with the atmospheric constituents. The absorbed energy of the electrons would reradiate without the presence of the neutral atmosphere. However, the electrons lose their energy due to the collisions with neutral particles which cause reduction of their reemitted signal."

4. Please clarify the reason of the analysis of the critical frequencies foE and foF2 (Fig. 1,2) while the absorption is the main subject of the paper.

Thank you for the question. Although the main subject of our manuscript is the absorption we wanted to show the behaviour of the ionospheric layers during the selected periods, too. However, based on your question and the comment of the other reviewer the Fig. 1., 2., 3. and their description can confuse the reader. Therefore, we deleted the first three Figures and their descriptions and we added a figure what shows a sequence of ionograms measured at two stations during the most intense flare events of our study (Fig. 1. in the revised manuscript). We hope that it helps to follow the behaviour of the ionosphere during this intense solar event. Furthermore, it makes clear the observation of total and partial radio fade-out and of fmin parameter at stations under different solar zenith angle what is the crucial part of our study.

We added the description of the Fig. 1. (in the revised manuscript) to the text as follows: "Here we demonstrate in detail the ionospheric response to an intense X17-class eruption that occurred on 28 October 2003. The European and South African ionosonde stations were located in the sunlit hemisphere during this flare event. Fig.1 shows a sequence of ionograms recorded close to the equator (Ascension Island) and at mid-latitude (San Vito) from 09:00 UTC to 14:30 UTC on 28 October 2003. Ionograms measured every 15 min were available for the analysis, however we show the records with 30 minute time resolution to cover the whole time interval of the flare from the start until the end of decay. The upper panel of Fig. 2 shows the X-ray variation between 06 (UTC) and 18 (UTC) recorded by GOES12 satellite. In the X-ray flux we can clearly observe the flare event that started at 09:51, reached its peak at 11:10 and ended at 11:24. The most directly observed ionospheric effect due to the X-class solar flare is the total and partial fade-out of the sounding HF waves on the ionograms (Fig. 1.). The disappearance of the traces caused by the enhanced ionospheric absorption was recorded at both stations. However, the duration of the total fade-out measured at the two observation sites was different. We may notice that an increase in the fmin

parameter was first detected in the ionogram at 10:00 (UTC) over Ascension Island, close to the dip equator (fmin increased to 5.4 MHz). At San Vito, located in southern Italy at mid-latitude, the effect was weaker at this time (fmin 2.9 MHz). The total attenuation of the radio waves was first recorded at Ascension Island at 11:00 (UTC). In the subsequent ionograms at 11:15 UTC (not shown here) and at 11:30 the total blackout was observed at both stations which coincided with the peak in the X-ray flux as it is shown at the upper panel in Fig. 2. The trace of the F region appears on the ionogram at San Vito at 12:00 (UTC), while the total radio fade-out remains at Ascension Island until 12:30 (UTC). With the decay in the X-ray flux the blackout became partial at both stations. The fmin parameter returns to its regular daily value ( 2.3 MHz) at San Vito at 14:00. The recovery over Ascension occurs later, partial radio fade-out was still detected at 14:30. We believe that the different duration of the total radio fade-out recorded in the ionograms at the two stations can be explained by the different solar zenith angle at the two sites. Since the degree of the radio wave absorption in the ionosphere varies with the solar zenith angle, we compared ionograms measured at stations under different solar zenith angles to research into the solar zenith angle dependence of the ionospheric response."

5. As I understand "The aim of the present study is the investigation of the solar flare effects on ionospheric absorption at mid- and low-latitudes taking into account the solar zenith angle: : :.". I would like once again to draw you attention to the D-RAP model (which is definitely based on the solar zenith angle dependence). I think that this model should be mention in your paper. The correlation between your results and D-RAP model results should be discussed.

Based on our results the solar zenith angle has to be take into account in the models describing the absorption of the ionosphere, like in the D-RAP model. We wrote few sentences about the importance of the solar zenith angle and the D-RAP model in the discussion part as well:

"Our results are in agreement with D-RAP model

(https://www.swpc.noaa.gov/products/d-region-absorption-predictions-d-rap/) on the dependence of solar zenith angle. This model was developed based on the theoretical descriptions of the ionospheric absorption by Davies (1990) and Sauer and Wilkinson (2008). According to the model the Highest Affected Frequency (HAF) is largest at the sub-solar point and it decreases with increasing solar zenith angle." ... "Therefore, our observations confirm the results of Zhang and Xiao (2005), Spirathi et al. (2013) and the D-RAP model that the solar zenith angle plays an important role in the ionospheric response to solar flares."

6. I suppose that the number of references to other authors in section "Results" could be diminished.

Thank you. We deleted them from there and discuss their results in comparison with our findings in the discussion part.

7. Table 2. Please, mention in the caption the unit ("UT", probably).

Thank you, we added UTC to the header of the table.

8. Table 3 and 4. Please, add the units in the titles "Solar zenith angle", "Duration of fade-out", "fmin" and "dfmin".

Thank you, we added the units to the header of the table.

9. I suppose that the values of the fmin have to be presented with equal (and reasonable) precision (perhaps one decimal point) in the Tables 3 and 4. The similar issue with dfmin.

We agree with your suggestion and changed the values in Table 3 and 4.

10. Figure 1. I thing it is better to write "first panel", "second panel" etc., instead of "upper plot" and "second upper plot"

We changed the word "plot" to "panel" in the figures's captions.

11. It seems to me that axes labels size and titles font size in the Figures 2-5 looks like very small.

Thank you. The labels and titles of the figures (Fig. 1-3 in the revised manuscript) have been increased in order to be more readable.

Please also note the supplement to this comment:
https://www.ann-geophys-discuss.net/angeo-2019-14/angeo-2019-14-AC1-supplement.pdf
* * *
**San Vito - 2003 - 10 - 28**

**Ascension Island - 2003 - 10 - 28**

Height [km]

Frequency [MHz]

**Fig. 1.**

---

## Author Comment (AC2) · 18 Apr 2019

Comments on the manuscript Angeo-2019-14

'Investigation of the ionospheric absorption response to flare events during the solar cycle 23 as seen by European and South African ionosondes' Submitted to Annales Geophysicae By Veronika Bartaet al.

General Comments: This work shows an analysis of ionospheric parameters in midand low-latitudes in relation to solar flares occurred in solar cycle 23. The authors inves-

ANGEOD

tigated the radio wave absorption in D layer, in which they defined a dfmin parameter as a good qualitative measurement to analyze this absorption. They show an interest analysis with interesting results. However, the authors needs to organize the results and deepen in the physical discussions. Therefore, the authors need to improve significant modifications. This paper needs a major revision. Furhermore, the authors need to improve English significantly.

We would like to thank the work of Reviewer #2 and their advices. We took into account them and we refined the text of the manuscript based on their comments (as it will be listed below). We made changes and additions throughout the text, mostly in the introduction, results and discussion. A few more references have also been added based on the Reviewer's suggestion. We tried to correct the typos and mistakes and improve the English of the whole manuscript.

We hope that the revised paper will now meet with the referee's approval. The changes in the manuscript which have been performed based on the second referee's questions/comments are indicated by red.

Major Comments: 1. Abstract: The abstract is not well written. I do not understand the main objective of this study. There are some typo English mistakes as "mimimum", "ionosopheric". The authors need to clarify better the purpose of this work.

Thank you for the comment. The first part of the abstract has been rewritten taking into account your suggestions. In the first sentence we tried to clarify better the purpose of our study. Furthermore, the typos have been corrected. The revised abstract is the following:

"We have investigated the solar flare effects on ionospheric absorption with the systematic analysis of ionograms measured at mid- and low-latitude ionosonde stations under different solar zenith angles. The lowest recorded ionosonde echo, the minimum frequency (fmin, a qualitative proxy for the "nondeviative" radio wave absorption occurring in the D-layer), furthermore and the dfmin parameter (difference between the
value of the fmin and the mean fmin for reference days) have been considered. Data was provided by at meridionally distributed ionosonde stations in Europe and South Africa during eight X and M class solar flares in solar cycle 23. Total and partial radio fade-out was experienced at every ionospheric stations during intense solar flares (> M6). The duration of the total radio fade-out varied between 15 and 150 min and it was highly dependent on the solar zenith angle of the ionospheric stations. Furthermore, a solar zenith angle-dependent enhancement of the fmin (2-9 MHz) and dfmin (1-8 MHz) parameters was observed at almost every stations. The fmin and dfmin parameters show an increasing trend with the enhancement of the X-ray flux. Based on the our results, the dfmin parameter is a good qualitative measure for the relative variation of the "nondeviative" absorption especially in the case of the less intense solar flares which do not cause total radio fade-out in the ionosphere (class < M6)."

2 Introduction (pag. 2, line 25): The solar flares cause an extra ionization in the D region, which causes an absorption of the HF waves, impairing the visualization of the E region in the data (ionograms, for example), and partially or totally in the F region. The authors affirm that there is an absorption in the E region, also. Please, clarify this part.

During a solar flare event, a great enhancement in extreme ultraviolet (EUV) and X-ray radiation causes increases in the ionospheric electron density not only in the D but also in the E and F regions (Tsurutani et al., 2005; Nogueira et al., 2015;). The electron collision frequency is highest in the D region ($2 \times 106$ s-1) and the HF radio waves below 10 MHz can be attenuated principally there (Zolesi and Cander, 2014). However, further studies have shown that solar flares can also cause enhancement of the neutral density and temperature of the thermosphere (Pawlowski and Ridley, 2008, 2011; Le et al., 2015). E. g. the model study by Pawlowski and Ridely (2008) has shown flare-induced density and temperature enhancements, with the effect decreasing from the 400 km (CHAMP satellite height) down to 110 km. According to the physical background of the ionospheric absorption the electrons accelerated by the electric field of

the transiting radio wave suffer collisions with the atmospheric constituents because of the presence of the neutral atmosphere and induce an energy loss which results in a reduction of their reemitted signal (Sauer and Wilkinson, 2008). Consequently, the enhanced neutral density and the temperature in response to solar flare increasing also the number of collisions thus the ionospheric absorption.

In the above mentioned part of the manuscript (Introduction (pag. 2, line 25):) we wrote the following: "The loss of HF communication as a consequence of the enhanced absorption affects navigation systems, especially commercial aircraft operations. Thus the monitoring of the absorption and D-, E-region electron density variation is an important issue from a practical point of view as well." Therefore, we didn't write about the absorption occurring in the E region explicitly. We only stated that "...E-region electron density variation is an important issue..."

Nevertheless, in our study we focus on the ionospheric absorption variation in response to solar flares and not on the E region electron density variation. Thus, we changed the text of the manuscript (page 2, lines 23-25) as follows: "The loss of HF communication as a result of the enhanced absorption affects navigation systems, especially in commercial aircraft operations. Therefore, describing, modelling and monitoring of the ionospheric absorption is an important issue from a practical point of view as well."

References: • Le, H., Ren, Z., Liu, L., Chen, Y., and Zhang, H.: Global thermospheric disturbances induced by a solar flare: A modeling study, Earth Planet. Space, 67, 1–14, https://doi.org/10.1186/s40623-014-0166-y, 2015 • Nogueira, P. A. B., Souza, J. R., Abdu, M. A., Paes, R. R., Sousasantos, J., Marques, M. S., Bailey, G. J., Denardini, C. M., Batista, I. S., Takahashi, H., Cueva, R. Y. C., and Chen, S. S.: Modeling the equatorial and low-latitude ionospheric response to an intense X-class solar flare, J. Geophys. Res.-Space, 120, 3021–3032, https://doi.org/10.1002/2014JA020823, 2015 • Pawlowski, D. J. and Ridley, A. J.: Modeling the thermospheric response to solar flares, J. Geophys. Res., 113, A10309, https://doi.org/10.1029/2008JA013182, 2008. • Pawlowski, D. J. and Ridley, A. J.:

The effects of different solar flare characteristics on the global thermosphere, J. Atmos.-Terr. Phys., 73, 1840–1848, 2011.   Sauer, H. H., and Wilkinson, D. C.: Global mapping of ionospheric HF/VHF radio wave absorption due to solar energetic protons. Space Weather, 6(12)., 2008   Tsurutani, B. T., Judge, D. L., Guarnieri, F. L., Gangopadhyay, P., Jones, A. R., Nuttall, J., Zambon, G. A., Didkovsky, L., Mannucci, A. J., Iijima, B., Meier, R. R., Immel, T. J., Woods, T. N., Prasad, S., Floyd, L., Huba, J., Solomon, S. C., Straus, P., and Viereck, R.: The October 28, 2003 extreme EUV solar flare and resultant extreme ionospheric effects: Comparison to other Halloween events and the Bastille Day event, Geophys. Res. Lett., 32, L03S09, https://doi.org/10.1029/2004GL021475, 2005.   Zolesi, B. and Cander, L.: Ionospheric Prediction and Forecasting, Springer Geophysics, Springer Heidelberg New York Dordrecht London, DOI 10.1007/978-3-642-38430-1., 2014

3 Introduction (pag. 3, line 32): It is necessary to define the fmim parameter; fmim of the F region, E region or both regions? The definition in the section "Method and data" is not enough to understand this part. The authors mention only the discussions about the fmim to be the minimum frequency of ionosphere, but in results (form of the data), I believe that fim refers to the F region. Please, clarify this part

Thank you for the question. In our study we analyzed the "general" fmin parameter, the minimum frequency of the echo trace observed in the ionograms. During our analysis we examined day-time ionograms, so generally the fmin should be the fmin of the E region. However, an enhancement in the fmin parameter can be occurred as a consequence of the increased D region radio wave absorption (see e.g. in study of Nogueira et al., 2015). In this case the first echo can be from the F region, consequently the fmin is fmin of the F region. To clarify it, we completed this part with the following sentence: "The fmin represents the minimum frequency of the echo trace observed in the ionogram and it is a rough measure of the nondeviative absorption (e.g. Davies, 1990)." Furthermore, we present a sequence of ionograms in Fig. 1. in the revised manuscript (as it was seen in previous papers e.g. Sahai et al. 2008, Nogueira et al.,

2015, Denardini et al. 2016.) providing the possibility to follow the variation of the fmin before and after the total radio fadeout.

4. Results: The results are interesting. Although this absorption is well known in the ionospheric data (Denardini et al, 2017, doi: 10,116/s40623-016-0456-7, Sahai et al., cited by authors, and other authors), the relation with the solar zenith angle is present in different form.

Thank you for the suggested papers. We read them carefully and wrote the most important findings into the introduction part. The text that has been added to the manuscript is the following:

"Solar flare effects on the equatorial and low- latitude ionosphere have been described by Sripathi et al 2013. They observed the lack of ionospheric traces in the ionograms during an X class solar flare and a strong blanketing type Es layer before and after the flare event. The total radio fade-out in the ionograms was observed simultaneously with an amplified signal amplitude in ground based VLF records. They suggested that the reason of the amplified VLF signals could be enhanced D region ionization due to solar flare which could also cause the increased absorption of HF radio waves observed in the ionograms. Partial radio fade-out and a blanketing type sporadic E layer were also detected in ionograms measured close to the equator in the Brazilian sector (Denardini et al. 2016). They determined a 42-146 % enhancement in the electron density of the E-layer after X-class solar flares with the observation of peaks in the fbEs parameter. The attenuation of radio waves (below 5–8 MHz) caused by ionospheric absorption occurred some minutes before the abnormal changes in the E region electron density and can be attributed to the additional X-ray ionization due to solar flares. Total radio blackout for about 70 min and increased values of the fmin parameter inferred from ionograms registered at two ionosonde stations in the equatorial region have been reported by Nogueira et al. (2015). The onset and recovery of the flare effect were observed with a consistent time difference at the two stations. Nogueira et al. (2015) stated that the reason for this time delay is the east-west separation of the observing

sites." ... "Nogueira et al. (2015) observed an abrupt increase of the TEC in the sunlit hemisphere due to a flare event. The plasma density perturbation seems larger and remains for longer time in the crest region of the equatorial ionization anomaly (EIA) than at the subsolar point. However, Spirathi et al (2013) demonstrated a good correlation between the TEC enhancement caused by a solar flare and solar zenith angle. This result verifies the study of Zhang and Xiao (2005) who have shown that the TEC varies with solar zenith angle."

However, the results are arranged in numerous figures and presented with a confusing text. It would be better to present the figures together (for example Figures 1 and 2 are a single figure)

Thank you for the suggestion. We changed the structure of the results part to make it clear and more readable. In the first paragraph we determined the particular issue of research: "In the present study we investigated the response of ionospheric absorption to solar flares with particular interest of the solar zenith angle dependence variation of it. We used ionograms measured at ionosonde stations under different solar zenith angle for the analysis. We calculated the solar zenith angles of the stations at the time of the peak of the 8 flares for the analysis. We examined three parameters that can be determined from ionograms: duration of the total radio fade-out, the value of the fmin parameter and the value of the dfmin parameter. In the first step we analyzed how the duration of the fade-out during the flare event depended on the solar zenith angle (Sec. 3.1). Secondly the solar zenith angle dependence of the fmin and dfmin parameters measured just after the fade-out were investigated (Sec 3.2). Then we repeated the analysis for the fmin and dfmin parameters measured at a certain time after the fade-out when we again recorded them at all the stations (3.3). In the last step the impact of the intensity variation on the absorption has been considered (3.4)."

Based on your comment and question/comments of the other reviewer the Fig. 1., 2., 3. and their description can confuse the reader. Therefore, we deleted the first three Figures and their descriptions and we added a figure what shows a sequence of

ionograms measured at two stations during the most intense flare events of our study (Fig. 1. in the revised manuscript). We hope that it helps to follow the behaviour of the ionosphere during this intense solar event. Furthermore, it makes clear the observation of total and partial radio fade-out and of fmin parameter at stations under different solar zenith angle what is the crucial part of our study.

We added the description of the Fig. 1. (in the revised manuscript) to the text as follows: "Here we demonstrate in detail the ionospheric response to an intense X17-class eruption that occurred on 28 October 2003. The European and South African ionosonde stations were located in the sunlit hemisphere during this flare event. Fig.1 shows a sequence of ionograms recorded close to the equator (Ascension Island) and at mid-latitude (San Vito) from 09:00 UTC to 14:30 UTC on 28 October 2003. Ionograms measured every 15 min were available for the analysis, however we show the records with 30 minute time resolution to cover the whole time interval of the flare from the start until the end of decay. The upper panel of Fig. 2 shows the X-ray variation between 06 (UTC) and 18 (UTC) recorded by GOES12 satellite. In the X-ray flux we can clearly observe the flare event that started at 09:51, reached its peak at 11:10 and ended at 11:24. The most directly observed ionospheric effect due to the X-class solar flare is the total and partial fade-out of the sounding HF waves on the ionograms (Fig. 1.). The disappearance of the traces caused by the enhanced ionospheric absorption was recorded at both stations. However, the duration of the total fade-out measured at the two observation sites was different. We may notice that an increase in the fmin parameter was first detected in the ionogram at 10:00 (UTC) over Ascension Island, close to the dip equator (fmin increased to 5.4 MHz). At San Vito, located in southern Italy at mid-latitude, the effect was weaker at this time (fmin $\sim$ 2.9 MHz). The total attenuation of the radio waves was first recorded at Ascension Island at 11:00 (UTC). In the subsequent ionograms at 11:15 UTC (not shown here) and at 11:30 the total blackout was observed at both stations which coincided with the peak in the X-ray flux as it is shown at the upper panel in Fig. 2. The trace of the F region appears on the ionogram at San Vito at 12:00 (UTC), while the total radio fade-out remains at Ascension

Island until 12:30 (UTC). With the decay in the X-ray flux the blackout became partial at both stations. The fmin parameter returns to its regular daily value ($\sim$ 2.3 MHz) at San Vito at 14:00. The recovery over Ascension occurs later, partial radio fade-out was still detected at 14:30. We believe that the different duration of the total radio fade-out recorded in the ionograms at the two stations can be explained by the different solar zenith angle at the two sites. Since the degree of the radio wave absorption in the ionosphere varies with the solar zenith angle, we compared ionograms measured at stations under different solar zenith angles to research into the solar zenith angle dependence of the ionospheric response."

5. Discussions and conclusions: The part of the discussion is actually a conclusion. The authors did not elaborate on the physical discussions. There are numerous studies about the subject of relation between flare solar and ionospheric parameters. I suggest that the authors to discuss further the results, that are very interesting, before being published in this journal.

Thank you for your suggestion. We read the papers carefully what you previously proposed in the review and compared our results with the most important findings of them. We believe that the more detailed discussion improved the quality of the manuscript. We must mention here that the coupled mechanisms in the magnetosphere-ionosphere-atmosphere system in response to solar flares are very complex but we focus on the changes of the ionospheric absorption and its solar zenith angle dependence in our study. Therefore, we discussed the results of previous papers only in connection with this topic.

We added the following parts to the discussion: "Total and partial radio fade-out were experienced at every ionospheric station during and after the X class solar flares (on 2001-09-24, 2003-10-28, and on 2005-12-05) and also in the case of some M class flares (e. g. on 2006-12-06). The observed time of the absence of the echoes was between 15 min and 150 min, similar to the findings of Sahai et al. (2006) with ionosondes over the Brazilian sector on 28 October 2003. Similarly, Nogueira et al. (2015)

found from a total to partial HF blackout for about 70 min in ionograms measured at the São Luís and Fortaleza equatorial stations as a result of an X2.8 solar flare. They observed a consistent time difference in the beginning and the end of the flare effect in the sequences of ionograms and they explained this phenomenon by the east-west separation of the observing sites. We investigated the beginning and the end of the total radio fade-out measured at the eastern locations as compared to the western locations. E.g. comparing the beginning and the end of the blackout at Chilton (west) with Juliusruh (east) or at Ascension Island (west) with Grahamstown (east) during the X17 flare occurring on 28 October 2003 (Fig. 2.) we cannot detect a systematic delay. Based on our results there is no detected east-west separated consistent time difference of the flare effect. Whereas, examining the duration of the total radio fade-out at the time of the same flare (28 October 2003, Fig. 2.) it seems to depend on the solar zenith angle. The smaller the zenith angle of the observation site (Grahamstown, Ascension Island) the longer the detected blackout of the HF waves. We observed a similar trend for the flares occurring on 05 December, 2006 and on 06 December, 2006 (Fig. 4.). The total radio fade-out during the time of intense solar flares (M > 5) could be understood due to absorption of radio signals by enhanced D region ionization. Previous studies reported that enhanced ionization of the D region can lower the reflection height of the VLF radio waveguide and amplify the amplitude of the propagating signals (Thomson and Clilverd, 2001; Thomson et al., 2004; Kolarski and Grubor, 2014). Sripathi et al 2013 observed lack of ionospheric traces in the ionograms simultaneously with an amplified amplitude signal of ground based VLF records during an X class solar flare. Their results suggest there could be enhanced D region ionization due to solar flare which also caused absorption of HF radio waves in the ionograms." . . . "Contradictory results have been reported in the literature about the solar zenith angle dependence of the ionospheric response to solar flares. Our results are in agreement with D-RAP model (https://www.swpc.noaa.gov/products/d-region-absorption-predictions-d-rap/) on the dependence of solar zenith angle. This model was developed based on the theoretical descriptions of the ionospheric absorp-

tion by Davies (1990) and Sauer and Wilkinson (2008). According to the model the Highest Affected Frequency (HAF) is largest at the sub-solar point and it decreases with increasing solar zenith angle. Moreover, Zhang and Xiao (2005) and Spirathi et al. (2013) have demonstrated a good correlation between the TEC enhancement caused by solar flares and the solar zenith angle, too. However, Li et al. (2018) concluded that there is no strong relationship between the Ne variation of the D region and the solar zenith angle. Furthermore, Nogueira et al. (2015) demonstrated an abrupt increase of the TEC. The observed anomaly seemed larger and remained for a longer time in the crest region of the equatorial ionization anomaly (EIA) than at the subsolar point. We also observed the largest and the longest-lasting perturbation of the ionospheric absorption in the equatorial region (at Ascension Island) in most of the cases. However, our results suggest that the solar zenith angle of the observation site plays an important role. For instance, at the peak time of the X9 flare (05 December 2006) the zenith angle of the ionosonde station at Ascension Island (geomagnetic latitude: -2.31°) was 36.14° and the duration of the fade-out was 60 min, smaller than measured at Grahamstown (geomagnetic latitude: -34.01°, see Table 3.). Even a larger difference was observed at the two stations during the M5-class flare at 09:27 on 27 October 2003. The solar zenith angle of Ascension Island was 47.96° at the peak time and there was no detected total radio fade-out. While at Grahamstown with a smaller solar zenith angle (21.77°) the duration of the total attenuation of HF waves was 150 min (Table 3.). Therefore, our observations confirm the results of Zhang and Xiao (2005), Spirathi et al. (2013) and the D-RAP model that the solar zenith angle plays an important role in the ionospheric response to solar flares."

Minor Comments: • English needs to improve in all manuscript: grammar, typo mistakes, absence of commas, and verbal agreement.

We tried to correct the typos and mistakes and improve the English of the whole manuscript.

• Legend of the figures (1 up to 5) are very difficult to see. Thank you. The labels

and titles of the figures (Fig. 1-3 in the revised manuscript) have been increased in order to be more readable.

Please also note the supplement to this comment:
https://www.ann-geophys-discuss.net/angeo-2019-14/angeo-2019-14-AC2-supplement.pdf

———————————————————

[Figure]

[Figure]

**Fig. 1.**

---

## Referee Report (RR1)

Comments on the manuscript

Angeo-2019-14

**'Investigation of the ionospheric absorption response to flare events during the**

**solar cycle 23 as seen by**

**European and South African ionosondes''**

Submitted to

**Annales Geophysicae**

By

**Veronika Bartaet al.**

**General Comments:**

The paper had significant improvements. The English, the figures and also the physical discussions were included and improved. It is an important work with good results. However, I still have few questions and suggestions before the publication. After the authors improve this part, I recommend for publishing.

**Comments:**

1. **Introduction**: The authors commented on Sripathi et al 2013 work (lines 20-22). The strong Es layers that occurred before of the solar flare were due to the wind shear mechanism presence during the Counter Electrojet (CEJ) event. Therefore, the Es layers occurrence does not related to the flare event. The CEJ event that is related to the solar flare, and in turn, the blanketing Es layers occurred in equatorial regions. Please, consider to explain with more details about this work or remove this discussion.

2. The authors need to explain Figure 1 with more detail. I did not understand the black line in the figure. I imagine it to be the fmim parameter. Authors need to better to define this parameter in the manuscript since it is using a general criterion. I would also like to have a more little discussion regarding the differences in the equatorial region and mid-latitudes in relation to the blackout events that occurred in this day.

3. **Discussion**: Line 27-30 -First paragraph is very confusing after the modifications.

4. Li et al is missing is missing an end point (page 13).

---

## Author Response (AR2)

**Response to reviewer 1.:**

**Comments on manuscript** "Investigation of the ionospheric absorption response to flare events during the solar cycle 23 as seen by European and South African ionosondes"

**Suggestions for revision or reasons for rejection** (will be published if the paper is accepted for final publication)
The manuscript was mostly corrected by authors but still needs an additional minor revision.

We glad to hear that according to the reviewer's opinion "the manuscript was mostly corrected" and we would like thank for their further suggestions to improve the quality of our manuscript. We addressed all their points by providing changes/additions in the revised manuscript, and responses here to their comments.

We hope that the revised paper will now meet with the referee's approval. The changes in the manuscript which have been performed based on the first referee's questions/comments are indicated by pink.

1)      I propose to add in Introduction the section with description of the fmin method (idea of the method, its advantages, limitations and different modifications).

We added a paragraph about the fmin method into the introduction part, as follows:
"The minimum frequency of reflection in radio soundings by ionosondes (fmin, Fig. 1.) is usually considered as a qualitative measure of the ''nondeviative'' radio wave absorption in the ionosphere (Risbeth and Gariott, 1969, Davies, 1990). The basis of the so called "fmin method" is to use this parameter as an absorption index during periods of high absorption which occur e. g. at the time of solar X-ray flares and polar cap absorptions (see Ch. 7. in Davies, 1990). Since the nondeviative absorption varies inversely as the square of the radio frequency, when the absorption changes are large there is a low-frequency cutoff on ionograms (fmin) which is roughly a function of ionospheric absorption for a given sounding system (Davies, 1990). However, the value of the absolute absorption occurring in the ionosphere can not be quantitatively determined from the fmin parameter. It is regularly used to investigate the absorption variation of the D region causing by geomagnetic storms (Oksman et al., 1981), by planetary waves (Schmitter et al., 2011) or by other effects (Kokourov, 2006)."
Furthermore, we deleted the general description of the fmin method from the Method and data part since we inserted it into the introduction.

Moreover, we added a Figure which shows a sample of ionogram (Fig. 1. in the revised2 manuscript). We indicated the fmin parameter on the ionogram.

2)      Caption Figure 1: Please, add the reference concerning source of the ionograms. I suppose it is GIRO site, but files downloaded from GIRO network [http://giro.uml.edu/] look slightly different.

Yes, the source of the ionograms is the GIRO network, but they have been cuted and postprocessed in order to show the sequence of ionograms for the whole period.
Nevertheless, we added the following sentence into the caption of Fig1. and Fig2. in the revised2 manuscript:
"Source of the ionogram is the GIRO network: Global Ionospheric Radio Observatory (GIRO, http://giro.uml.edu)"

3)      Figures 5-8: It seems to me that it is possible to combine them (especially fmin and dfmin) for better understanding.

Thank you for the advice. We tried to combine the plots about fmin and dfmin solar zenith angle dependence during the flare event 28, October 2003 into one Figure. We insert it here. It does not seem to help the understanding. We afraid that it even became more confusing for the reader because he/she can mix up the point of the two variable. Therefore, we prefer to plot the fmin and dfmin parameters separately for the individual events as it was the case in the previous manuscript.

[Figure]

4)      Caption Figure 4: I think that preposition "ON" better be used only before first data "on 28 October 2003 (a), 5 December 2006 (b), 24 September 2001 (c), and 6 December 2006 (d)."

Thank you. We corrected it.

5)      "Investigation of the ionospheric absorption response to flare events during the solar cycle 23 as seen by European and South African ionosondes":

10   It seems to me that the title of the paper does not reflect the contents of the article.

In reality, you investigate the parameters of the ionograms but not absorption. You even did not try to assess the values of the absorption.

Maybe something similar:

"Investigation of the ionogram dynamics during flare events at the solar cycle 23 as seen by European and South African

15   ionosondes".

Thank you for your suggestion. We agree with that. We changed the title of the manuscript to the following:
"Effects of solar flares on the ionosphere as shown by the dynamics of ionograms recorded in Europe and South Africa"

**Response to reviewer 2.:**

Comments on the manuscript
Angeo-2019-14

'Investigation of the ionospheric absorption response to flare events during the solar cycle 23 as seen by European and South African ionosondes''

Submitted to
Annales Geophysicae
By
Veronika Bartaet al.

**General Comments**:
The paper had significant improvements. The English, the figures and also the physical discussions were included and improved. It is an important work with good results. However, I still have few questions and suggestions before the publication. After the authors improve this part, I recommend for publishing.

Thank you for the reviewer his/her positive opinion in connection with the improvement of the manuscript and also for her/his further suggestions/comments. We addressed all their points by providing changes/additions in the revised manuscript, and responses here to their comments.

We hope that the revised paper will now meet with the referee's approval. The changes in the manuscript which have been performed based on the first referee's questions/comments are indicated by red.

Comments:
1.      Introduction: The authors commented on Sripathi et al 2013 work (lines 20-22). The strong Es layers that occurred before of the solar flare were due to the wind shear mechanism presence during the Counter Electrojet (CEJ) event. Therefore, the Es layers occurrence does not related to the flare event. The CEJ event that is related to the solar flare, and in turn, the blanketing Es layers occurred in equatorial regions. Please, consider to explain with more details about this work or remove this discussion.

Thank you for the reviewer that he/she called our attention of this mistake. We removed this discussion from the manuscript.

2.      The authors need to explain Figure 1 with more detail. I did not understand the black line in the figure. I imagine it to be the fmim parameter. Authors need to better to define this parameter in the manuscript since it is using a general criterion. I would also like to have a more little discussion regarding the differences in the equatorial region and mid-latitudes in relation to the blackout events that occurred in this day.

Thank you for your advice. Based on your and the other reviewer's suggestions we inserted the following paragraph about the fmin parameter and the fmin method:
"The minimum frequency of reflection in radio soundings by ionosondes (fmin, Fig. 1.) is usually considered as a qualitative measure of the ''nondeviative'' radio wave absorption in the ionosphere (Risbeth and Gariott, 1969, Davies, 1990). The basis of the so called "fmin method" is to use this parameter as an absorption index during periods of high absorption which occur e. g. at the time of solar X-ray flares and polar cap absorptions (see Ch. 7. in Davies, 1990). Since the nondeviative absorption

varies inversely as the square of the radio frequency, when the absorption changes are large there is a low-frequency cutoff on ionograms (fmin) which is roughly a function of ionospheric absorption for a given sounding system (Davies, 1990). However, the value of the absolute absorption occurring in the ionosphere can not be quantitatively determined from the fmin parameter. It is regularly used to investigate the absorption variation of the D region causing by geomagnetic storms (Oksman et al., 1981), by planetary waves (Schmitter et al., 2011) or by other effects (Kokourov, 2006)."

Moreover, we added a Figure which shows a sample of ionogram (Fig. 1. in the revised2 manuscript). We indicated the fmin parameter on the ionogram with a vertical black line. We hope so that after this paragraph and the sample ionogram the Fig. 2. (sequence of ionograms on 28 October, 2003) is more clear. The vertical black lines show the fmin parameter, like on the sample ionogram. Furthermore, we completed the caption of the Fig. 2. as follows:

"The black vertical lines show the fmin parameter on the ionograms (like on the sample ionogram in Fig. 1.)."

We completed also the description of the Fig. 2. (Fig. 1. in the earlier manuscript) in the text. It is the following in the new manuscript:

The most directly observed ionospheric effect due to the X-class solar flare is the total and partial fade-out of the sounding HF waves on the ionograms (Fig. 2.). The disappearance of the traces caused by the enhanced ionospheric absorption was recorded at both stations. However, the duration of the total fade-out measured at the two observation sites was different. We may notice that an increase in the fmin parameter (marked by the vertical black lines on Fig. 2.) was first detected in the ionogram at 10:00 (UTC) over Ascension Island, close to the dip equator (fmin increased to 5.4 MHz). In contrary at San Vito, located in southern Italy at mid-latitude, the effect was weaker at this time (fmin ~ 2.9 MHz). It indicates, that the increased absorption, caused by the solar flare, has been earlier detected close to the equator than at mid-latitude. The total attenuation of the radio waves was first recorded at Ascension Island at 11:00 (UTC). In the subsequent ionograms at 11:15 UTC (not shown here) and at 11:30 the total blackout was observed at both stations which coincided with the peak in the X-ray flux as it is shown at the upper panel in Fig. 3. Therefore, the ionospheric absorption caused by the flare event had a maximum during this period. The trace of the F region appears on the ionogram at San Vito at 12:00 (UTC), while the total radio fade-out remains at Ascension Island until 12:30 (UTC). With the decay in the X-ray flux the blackout became partial at both stations. The fmin parameter returns to its regular daily value (~ 2.3 MHz) at San Vito at 14:00. It shows the end of the high absorption period caused by the flare at mid-latitude. The recovery over Ascension occurs later, partial radio fade-out was still detected at 14:30 indicating that the impact of the ionospheric absorption is still detectable at the equatorial region. We believe that the different duration of the total radio fade-out recorded in the ionograms at the two stations can be explained by the different solar zenith angle at the two sites. Based on the theoretical description (Davies, 1990) and model (D-RAP2) the degree of the radio wave absorption in the ionosphere varies with the solar zenith angle. Therefore, the absorption variation caused by the solar flare is largest at the subsolar point (solar zenith angle = 0) and it decreasing with increasing solar zenith angle. Thus, the period of the total radio fade-out caused by the increased absorption should be longer close to the equator than at mid-latitude (as it can be observed in Fig. 2.). In order to investigate the solar zenith angle dependence of the ionospheric response we compared ionograms measured at stations under different solar zenith angles at the time of the flare.

3.      Discussion: Line 27-30 -First paragraph is very confusing after the modifications.

Thank you. We changed the text as follows:

" The solar flare effects on ionospheric absorption at mid- and low-latitude have been investigated with the systematic analysis of ionograms during eight X and M class flares."

4.      Li et al is missing is missing an end point (page 13).

Thank you, we corrected it. Furthermore, we checked the whole manuscript and we found this problem in other places as well, we corrected everywhere.
* * *

[revised manuscript text omitted]